# PsyDI: Towards a Personalized and Progressively In-depth Chatbot for Psychological Measurements

## Abstract

In the field of psychology, traditional assessment methods, such as standardized scales, are frequently critiqued for their static nature, lack of personalization, and reduced participant engagement, while counseling evaluations are often inaccessible to the general public. The complexity of quantifying psychological traits further limits these methods. Despite advances with large language models (LLMs), many still depend on single-round Question-and-Answer interactions. To bridge this gap, we introduce *PsyDI*, a personalized and progressively in-depth chatbot designed for psychological measurements, exemplified by its application in the Myers-Briggs Type Indicator (MBTI) framework. *PsyDI* leverages user-related multi-modal information and engages in customized, multi-turn interactions to provide personalized, easily accessible measurements, while ensuring precise MBTI type determination. To address the challenge of unquantifiable psychological traits, we introduce a novel training paradigm that involves learning the ranking of proxy variables associated with these traits, culminating in a robust score model for MBTI measurements. The score model enables *PsyDI* to conduct comprehensive and precise measurements through multi-turn interactions within a unified estimation context. Through various experiments, we validate the efficacy of both the score model and the *PsyDI* pipeline, demonstrating its potential to serve as a general framework for psychological measurements. Furthermore, the online deployment of *PsyDI* has garnered substantial user engagement, with over 3,000 visits, resulting in the collection of numerous multi-turn dialogues annotated with MBTI types, which facilitates further research.

## 1 Introduction

Recent progress in general-purpose foundation models, such as large language models (LLMs)(Anthropic, 2023; Touvron et al., 2023) and vision language models (VLMs)(Li et al., 2023; Team et al., 2023; Liu et al., 2024) has shown that artificial intelligence (AI) systems have capabilities to chat, reason, and incorporate relevant context to realize naturalistic interactions. Their advancements provide an opportunity to expand the forms of psychological measurements with AI and make it customized for each user. Traditional assessments often utilize self-report standardized scales and questionnaires for surveys and analysis (Cohen et al., 1996). While this approach can quickly cover a large population, it suffers from relatively weak data reliability and is unable to conduct targeted tests for specific individuals. On the other hand, professional psychological interviews and counseling are not easily accessible to the general public. Correspondingly, due to building upon billions of human language knowledge, an LLM-powered system can provide user-specific interaction experiences in the psychological dialogue. These agents have the potential to uncover the uncertainties on the surface and dig out the underlying information of different users, then summarizing and analyzing these information based on some professional psychological knowledge injected in prompts or fine-tuning data.

Several recent works attempt to combine neural network techniques with clinical knowledge to improve the performance and generalization on mental disease counseling (Toleubay et al., 2023) and emotional support (Buechel et al., 2018; Cheng et al., 2023) tasks. To enhance the richness of knowledge and the logical reasoning, various methods (Tu et al., 2024) have introduced LLMs, fine-tuning them with a variety of specialized psychological data to make them more suitable for these scenarios. Other approaches (Yang et al., 2024) have adopted role-playing dialogues, guiding LLMs to provide a more immersive psychological dialogue experience. And some of these methods offer an online chat entry for trial, significantly lowering the barrier to entry and promoting wider adoption among the population.

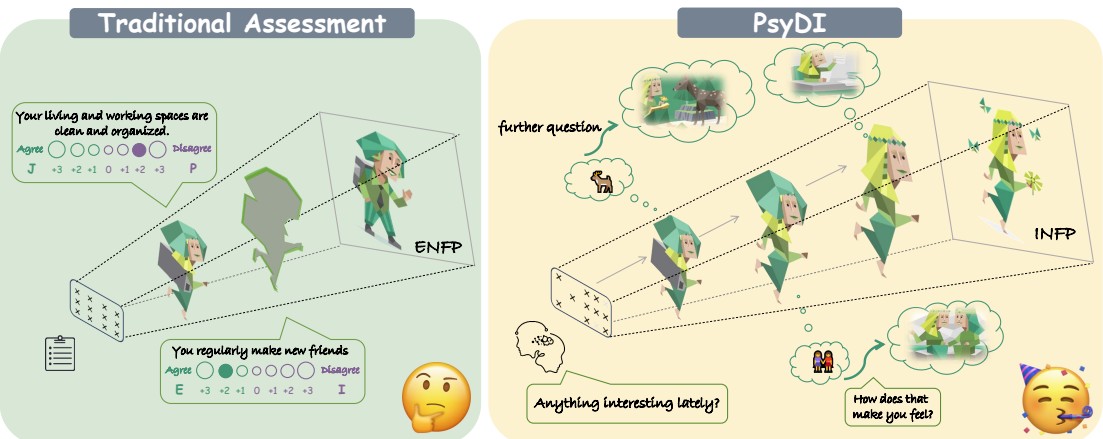

Figure 1: Comparison of traditional psychological assessment and the *PsyDI* framework. **(left)** Traditional assessment, which relies on general questions, can only discern external behavioral traits of the user. This often leads to erroneous MBTI measurements due to controversial external behaviors, such as the INFP person exhibiting some ENFP behaviors. **(right)** In contrast, the *PsyDI* framework poses questions based on familiar life scenarios, such as topics like friendship or workplace, and gradually delves into the user's internal cognitive functions through their external behaviors. This comprehensive approach allows for a more accurate identification of the underlying causes of controversial behaviors and provides a precise MBTI measurement.

Despite these advancements, the existing works remain confined to the realm of single-turn textual question-and-answer (Nori et al., 2023) (Q&A). These methods struggle to maintain style and logic consistency across multi-turn dialogues and fail to offer accurate and stable quantitation measurements over multiple instances for the same user. In this paper, we undertake a rigorous investigation to devise a comprehensive framework that not only offers an engaging interactive experience but also delivers reliable quantifiable analysis outcomes (Figure 1). Our exploration starts from a formalization of psychological measurements within the context of multi-turn decision-making processes (Section 2). Specifically, psychological measurements with LLMs are conceived as a multi-turn interaction process designed to evaluate specific discrete types or continuous scores of the test-taker. These measurements should be presented in a manner that is both engaging and succinct, ensuring its acceptability to the test-taker. We encapsulate these considerations into the following principles: ① A superior psychological measurement should, akin to the meticulous approach of human experts in psychological typology, actively and ethically gather and construct the user portrait from multi-modal question-answering formats. ② It should represent a complex skill whose optimal result is highly dependent on the user context, including cultural, linguistic, and individual factors to minimize the biased impact. (Altarriba & Santiago-Rivera, 1994) ③ It should provide solutions for psychological traits that are difficult to quantify, while achieving results that are equivalent to or superior than those yielded by current scale-based assessments, with rigorous validation and standardization processes.

Based on the above discussed principle, we introduce a comprehensive psychological measurement framework named *PsyDI*, and demonstrate its effectiveness on one of the most popular personality measurement: Myers-Briggs Type Indicator (MBTI) (Myers et al., 1962). This framework are composed of two innovative designs: progressively in-depth pipeline and score model. Firstly, recognizing the scarcity of complete multi-turn MBTI measurement dialogue data, we deem it impractical to design an end-to-end dialogue system. Thus, we adopt a progressively multi-phase dialogue paradigm. Drawing inspiration from widely-used psychological scales, we also design an MBTI profile to select topics from user's statements for each phase and quantitatively monitor the user's MBTI predilection. This design leverages the powerful in-context learning abilities of LLMs to construct an engaging multi-turn chat within each phase, while selecting the dialogue topic and updating the MBTI profile at the start and the end of each phase, effectively decoupling these two components. Furthermore, to refine the updates of the profile, we propose a score model training technique, which ranks MBTI indicativeness by leveraging the accuracy of how well statements are predicted to align with a specific MBTI type as a proxy variable. Additionally, this technique employs a ranking loss function to enhance robustness against noise. We then detail a series of automated data generation scheme and neural network

architectures to train a score model, akin to the concept of preference-based reinforcement learning (Wirth et al., 2017). All these components are integrated into a multi-modal and interactive chatbot.

To demonstrate the effectiveness of this framework, we initially conduct several quantitative experiments and ablations about several key designs, including comparisons between various LLMs and our score models across diverse datasets (Section 5.2 and Section 5.3). Subsequently, we release an online accessible version of *PsyDI* and collect data from over 3,000 participants. Based on de-identified data, we perform a series of analysis and visualizations, examining both specific questions and the whole test process. These examinations validate the framework's proficiency in dissecting the underlying characteristics of users. Concurrently, it provides qualitative evidence of *PsyDI*'s capacity for continuous refinement in the measurement of the user's MBTI. Additionally, we extend *PsyDI* to the emotion analysis scenario to validate its transferability (Appendix E.4). The main contributions of this paper can be summarized as follows:

- We introduce *PsyDI*, a novel AI agent framework designed to transcend the limitations of static psychological measurements (e.g. MBTI) through a multi-modal, personalized, progressively in-depth pipeline.
- To enhance the validity and explanability of *PsyDI*, we propose a simple yet effective data generation schemes for training the score model, with a novel optimization method that implicitly estimates unmeasurable variables by learning the ranking of proxy variables.
- Experimental findings derived from quantitative metrics and qualitative real-world user tests, encompassing 3,000 participants, substantiate the potential of *PsyDI* as a general psychological interaction agent.

## 2   Problem Formulation

We initiate our analysis by modeling the measurement of psychological indicators in Section 2.1 based on the characteristics of MBTI, as detailed in the Appendix A. Then we introduce the relationship between the MDP modeling and our designed framework in Section 2.2

### 2.1   MDP Definition for psychological measurement

We conceptualize the interaction between LLM agents and the test taker (user) as a Markov Decision Process (MDP) (Sutton & Barto, 2018). This formulation allows us to systematically optimize the sequence of inquiries posed to the user, with the aim of enhancing the accuracy, efficiency and engagement of MBTI evaluations. Formally, the MDP is articulated as $\mathcal{M} = \{\mathcal{S}, \mathcal{A}, \mathcal{P}, \mathcal{R}, \gamma\}$. The states $s$ in the observable state space $\mathcal{S}$ are defined as $s = \{p_1, \ldots, p_n\}$, where $p_i$ is the $i$-th textual statement of the user (*e.g. I like to talk to different people and share my stories*). Additionally, the state space includes a special terminated state $s_{\text{exit}}$ representing the user's decision to exit the evaluation process.

Throughout the process, the agent needs to determine to either persistently delve deeper into user's cognitive preferences through multi-turn iterative Q&A, or to terminate the process and render a forecasted MBTI type of the user. We define MBTI as $m$, a four-dimensional vector composed of elements 1 and -1, which symbolizes the orientation (E/I, N/S, F/T, J/P) of the MBTI along its four dimensions. For instance, the MBTI type INFP is represented by the vector: $m = [-1, 1, 1, -1]$. While gathering more question-answers may narrow down the user's authentic MBTI, it is imperative to acknowledge that an extended engagement may engender user impatient. Therefore, the action space $\mathcal{A}$ is defined as the union of these two kinds of action set $\mathcal{A} = \mathcal{A}_q \cup \mathcal{A}_t$. Here each $a$ in $\mathcal{A}_q$ represents a textual question $a = \{w_1, \ldots, w_L\}$ and $\mathcal{A}_t$ represents the set of termination actions, i.e., all possible MBTI types, where $L$ is the length of the question. Once the user selects an answer, the question together with the answer can be regarded as a more detailed user statement $p'$. Consequently, the state transition for the state $s$ is $s' = s \oplus p'$, where $\oplus$ signifies the concatenation of the current state with the new statement $p'$. In the event that the user opts to exit, the state transmutes to $s_{\text{exit}}$.

The reward function, which maps $\mathcal{S} \times \mathcal{A}$ to $\mathbb{R}$, is designed to measure the similarity between the predicted MBTI and the user's authentic MBTI. Concurrently, it imposes a penalty for the duration of the interaction and the incidence of user withdrawal. Formally, the reward function is defined as:

$$r(s_t, a_t) = \begin{cases} -c, & a \in \mathcal{A}_q, s' = s_{\text{exit}} \\ -|a, m^*| - \lambda \cdot t, & a \in \mathcal{A}_t \end{cases} \tag{1}$$

where $|a_t, m^*|$ denotes the measure of similarity between the predicted and the true MBTI type $m^*$, $\lambda$ is the penalty factor for the cumulative interaction steps $t$ while $c$ is the penalty positive constant for the exit.

In summary, the MDP employed in this context is similar to challenges tackled in goal-conditioned reinforcement learning (GCRL) (Liu et al., 2022). The purpose of this decision-making process is to achieve a specific objective (goal), i.e., to accurately predict user's MBTI type within the minimal number of interaction steps.

## 2.2 Relationship between MDP and Practical Framework

The MDP environment previously defined is an idealized model, providing a clear framework for subsequent analysis. Realistically, we can hardly construct a perfectly ideal MDP due to the inability to directly obtain users' true psychological indicators, thus precluding accurate reward determination. Instead, we employed a heuristic algorithm to handle reward uncertainty flexibly.

The original action space in MDP we defined is $\mathcal{A} = \mathcal{A}_q \cup \mathcal{A}_t$, where $\mathcal{A}_q$ represents the entirety of possible questions that can be posed. Considering the vastness of the original action space, we propose a heuristic algorithm that leverages an intuitive understanding that *a well-formed question should revolve around the user's statements* (Hill et al., 1992). By decomposing the action space into two sequential phases—first selecting a statement and subsequently generating a question centered on that statement—we effectively reduce the dimensionality of the action space. Specifically, the decomposed action space is $\bigcup_{i=1}^{n}(a_{p_i} \times Q_{p_i})$, where $a_{p_i}$ represents for selecting statement $p_i$ to ask following question and $Q_{p_i}$ is the question space related to statement $p_i$.

Leveraging the powerful language capabilities and goal-oriented abilities of LLMs within short conversational rounds, we further extend the question-generating process, denoted as $q_{p_i}$, into a multi-turn dialogue. This multi-turn dialogue is a sub-process, where each question is based on the statement $p_i$ and the history of question-answer pairs $h_j$ within this multi-turn dialogue. Consequently, the action $q_{p_i j}$ represents the $j$-th question in the multi-turn dialogue centered around $p_i$. Therefore, the action space is $\bigcup_{i=1}^{n}(a_{p_i} \times \bigcup_{j=1}^{m} q_{p_i j})$. In the subsequent pipeline design, we will demonstrate how to step-by-step select actions within this action space to optimize the return.

# 3 PsyDI Progressively In-depth Pipeline

In this section, we introduce the *PsyDI* pipeline, designed to enhance psychological assessments through a conversational approach. By decomposing the chat process into multi-turn conversations focused on specific statement, *PsyDI* ensures that large language models (LLMs) stay on track with their instructions. Inspired by traditional psychometric practices, we introduce the MBTI Profile as a continuous context to guide LLM questioning, detailed in Section 3.1. This Profile dynamically updates based on user statements, ensuring the conversation remains psychologically insightful. Additionally, during the multi-turn questioning phase, questions are centered around these informative statements, and a hybrid response format combining multiple-choice and free-form answers enhances user engagement and flexibility, as detailed in Section 3.3. Additionally, a hybrid response format combining multiple-choice and free-form answers enhances user engagement and flexibility, detailed in Section 3.3. Through these integrated strategies, *PsyDI* aims to provide a robust framework for psychologically informed conversations. The entire workflow of PsyDI is illustrated in Figure 2. For more details on the pipeline, please refer to Appendix D.

## 3.1 MBTI Profile

Similar to the quantitative metrics used in psychological scales, we initially establish a kind of estimation of the user's MBTI type, derived by their statements. In practice, we adopt the psychology profile commonly used in psychological measurements (Watson & Clark, 1994). This component serves to document the user's MBTI predilections and subsequently directs other modules of *PsyDI* in determining the most efficacious strategy for conducting multi-turn dialogues. Specifically, we design an **MBTI Profile** that documents the individual scores for each trait under measurement. It can translate the user's statement into quantifiable indicators that accurately reflect their MBTI predilections, while ensuring uniformity and cross-user comparability.

Considering the non-independence among the four dimensions of the MBTI, we assign a predilection to each type, encapsulating the likelihood that the user aligns with that specific MBTI type. The advantage of this approach is that it allows us to independently model the impact of a statement on each type. Furthermore, as the profile is continually updated with new statements, the estimated predilections for all MBTI types are dynamically adjusted to accurately reflect the evolving understanding of the user's personality traits.

The state influences the likelihood of a user being classified under each MBTI type. This classification is based on the degree to which each statement reflects the characteristics of a specific MBTI type, a metric we refer to as *Indicativeness*. To quantify it, we train a Score Model $\mathcal{F}_m(\cdot)$ to estimate the indicativeness of a statement under MBTI $m$. For the definition and the entire training process of Score Model, please refer to Section 4. Then the value of a statement within an MBTI type is determined by its quantile point, reflecting its level of indicativeness within the overall distribution of statements associated with MBTI type $m$:

$$\mathcal{V}_m(p) = \frac{|\{p' \in P_m | \mathcal{F}_m(p') \leq \mathcal{F}_m(p)\}|}{|P_m|} \tag{2}$$

where $P$ is the entire set of all statements made by users with MBTI $m$. $\mathcal{F}_m(p)$ is the indicativeness of statement $p$ while $\mathcal{V}_m(p)$ represents the value. For example, a statement with a value of 0.75 represents that 75% of the statements with the same MBTI in the dataset has an indicativeness less than this statement.

**Initialization of MBTI Profile**: For this, we compute the average indicativeness of all known statements attributed to the user. This operation is performed for each MBTI, allowing us to construct the user's initial profile, where $s$ represents for state and $\bar{\mathcal{F}}_m(\cdot)$ is the average of the indicativeness of all the statements in $s$:

$$\mathcal{V}_m^0(s) = \frac{|\{p' \in P_m | \mathcal{F}_m(p') \leq \bar{\mathcal{F}}_m(s)\}|}{|P_m|} \tag{3}$$

**Update of MBTI Profile**: During the update of MBTI profile, the incorporation of new statements serves as a way to refine the accuracy of the MBTI profile. This update process implicitly accounts for the interdependencies among the MBTI types, as the scoring model, trained to recognize these relationships, updates the scores in response to new statement. The impact of each new statement on the profile is contingent upon the initial estimates; for instance, if a user's profile strongly favors INFP, additional statements may not significantly alter this inclination. Consequently, the update process involves a linear mapping function that adjusts the MBTI profile based on the indicativeness of the new statement relative to the existing profile:

$$\mathcal{V}_m(s \cup \{p_{new}\}) = \mathcal{V}_m(s) + f(\mathcal{V}_m(s)) \cdot \mathcal{V}_m(p_{new}) \tag{4}$$

where $p_{new}$ is the latest statement, while $f(\cdot)$ is the non-linear mapping function designed to judiciously regulate the growth weights in accordance based on the current profile.

## 3.2 Statement Selection

Following the update of the MBTI Profile, another key challenge emerges in eliciting more representative statement from users through subsequent questioning. Given the preliminary acquisition of user portrait statements, the next process can be divided into two distinct phases: statement selection and multi-turn dialogue with the user. The goal of the former lies in discerning which statements are most indicative of representative responses, thereby delineating the areas requiring in-depth exploration and further analysis. On the other hand, the objective of multi-turn dialogue is to engage users, encouraging them to provide more details in the extended question-answers and maintain their interest throughout the measurement.

To determine the most indicative statements, we should exclude those with minimal informational value and those that explicitly describe a specific MBTI type, as these do not warrant further inquiry. Instead, we should focus on statements that are ripe for exploration, particularly those that exhibit ambiguity across multiple MBTI dimensions. For instance, considering a user whose statements could align with either the INFP or ENFJ types. The next questioning should be designed to delve into the motivations underlying inconsistent dimensions. Thus, we design a strategy to select statements that are ambiguous yet highly representative of

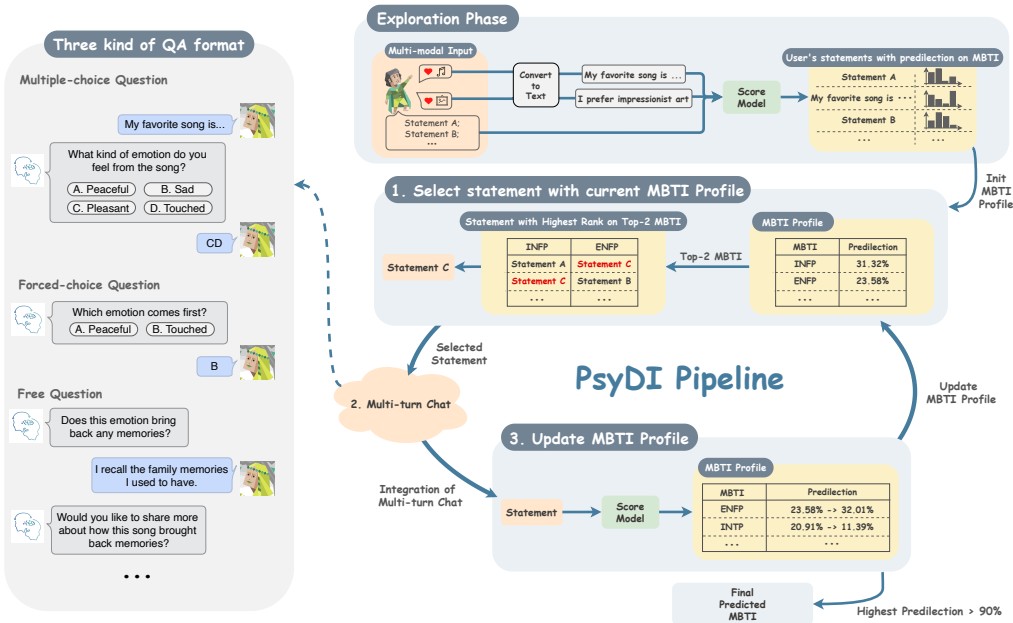

Figure 2: **Pipeline Overview**. *PsyDI* operates in a loop comprising three phases. The process begins with the user providing statements to initialize the MBTI profile. Based on the current profile, *PsyDI* selects a specific statement and engages in a multi-turn dialogue with the user. The interaction outcomes are used to update the profile. This iterative loop persists until *PsyDI* achieves high confidence in the user's MBTI.

both the I/E and P/J. This approach aims to refine our comprehension of the nuanced meaning underlying statements and to enhance our insight into the user's MBTI. This process can be formulated as:

$$\arg\min_{p \in s} \sum_{i=1}^{4} \left( M_j^i \otimes M_k^i \right) \cdot \left( \mathrm{rank}_i(\mathcal{F}_{m_j}(p)) + \mathrm{rank}_i(\mathcal{F}_{m_k}(p)) \right) \tag{5}$$

The notations $m_j$ and $m_k$ refer to the top-2 MBTI types within the current MBTI profile. $M$ represents the boolean $(0/1)$ counterpart of $m$, where $M_j^i$ denotes the boolean $(0/1)$ value of $m_j$ on the $i$-th MBTI dimension. The symbol $\otimes$ means XOR operation, which is employed to discern the differing dimensions between these two MBTI types. While $\mathrm{rank}_i$ signifies the rank of the scores on the $i$-th MBTI dimension across all statements. The objective of Eq. 5 is to pinpoint statements that score highly on all dimensions of ambiguity, enabling subsequent questioning to elucidate the user's true cognitive preferences.

### 3.3 Multi-turn Chat

Although LLMs perform exceptionally well in single-round, goal-oriented dialogues, it encounters difficulties in maintaining a coherent focus around a single objective when faced with a dozen rounds of multiple questions. To address this limitation, we have devised a hybrid approach that leverages LLMs' strengths by structuring interactions into modules, each centered on a specific statement. Within these modules, their capabilities are employed to progressively delve into questions related to MBTI, ensuring a step-by-step deepening of the exploration. Specifically, each module begins with a statement from the user, and PsyDI asks 3-5 rounds of progressively deeper questions related to that statement. The user's responses are then summarized into a single, refined self-description. For example, if the initial statement is "I feel stressed at work," after 3-5 rounds of questioning about the sources of stress, the summarized statement might be: "I feel stressed at work when I cannot meet deadlines. When this happens, I often relieve the stress by reorganizing my plans." The summarization of each module's dialogue then contributes new insights to the pipeline, facilitating a cumulative and systematic advancement of the overall structure.

The key challenge *PsyDI* needs to resolve at this phase is how to guide users to reveal their underlying motivations while maintaining their interest in our measurement, thereby ensuring a high completion rate.

This requires our questions to be progressively layered and logically interconnected, also engaging users throughout the process and encouraging them to provide deeper insights into their psychological profiles.

For the first challenge, to lead users to reveal their underlying motivations, we utilize multiple-choice questions where each option corresponds to a specific cognitive preference. By presenting such options, users can select the one that best aligns with their own, thereby avoiding ambiguous expressions and implicitly guiding them to reflect on their cognitive tendencies. For questions requiring confirmation, *PsyDI* employs forced-choice questions, compelling users to identify their most likely cognitive preferences. Also, *PsyDI* accepts free-form answers, allowing users to provide responses beyond the preset options. The prompt for this phase is list in Appendix C. For the second challenge, which is to keep user's interesting. We have designed a multimodal input system that allows users to input their preferred music and images, which are then converted into text to form a user statement. Additionally, our questioning framework is designed with a three-phase thought chain. First, we analyze the user statement, then we infer the user's underlying thought patterns, and finally, we derive the next question based on these patterns. This approach ensures that the entire questioning process revolves around the user's thinking style, providing greater motivation for them to continue answering. For a clearer example of the full conversation, please refer to Figure 13, 14, 15, and 16.

## 4 Score Model

To guide the direction of multi-turn chat using the MBTI Profile, it is essential to first determine the current MBTI estimation of the user. Consequently, we need to train a score model to infer the user's MBTI from their statements. However, the psychological indicators embedded in these expressions are inherently difficult to observe, and existing datasets of psychological indicators are often noisy and biased. To address these challenges, we propose a method that employs a proxy variable to indirectly observe the original variable, as introduced in Section 4.1. By utilizing rank loss in Section 4.2, the score model focus on learning the rank of the proxy variable rather than its absolute values, thereby ensuring the robustness of score model's indirect learning of the original variable. We also introduce the pair-wise dataset construction method in Section 4.3. This method generates a dataset where each sample consists of a pair of statements and their corresponding MBTI labels, which is used to train the rank loss.

### 4.1 Ranking Training Paradigm

According to the existing research on MBTI (Boyle, 1995; Furnham & Crump, 2005) and available datasets, the MBTI indicativeness of a self-report statement remains an unmeasurable variable, attributable to the inherent ambiguities in language. For instance, the phrase "I easily become lost in fantasies" may align with an NP preference if the user emphasizes possibilities, yet shift towards an NJ inclination if the user focuses on long-term future considerations. To address this limitation, we propose employing the accuracy of MBTI prediction, that is, the degree to which the LLMs' predictions for a given statement correlate with the assigned labels. This accuracy serve as a *proxy variable*, effectively substituting for the unmeasurable statement indicativeness. The rationale behind this choice lies in the quantifiability and relevance of the prediction accuracy to the statement indicativeness. Notably, specific MBTI personality types may have distinctive patterns in the expression and perception of information within their statements. For example, individuals with an F preference might employ a higher frequency of emotional language, whereas those with a T preference may emphasize problem-solving solutions. By employing the prediction accuracy, we establish an observable metric connected to the statement indicativeness. Therefore, we can implicitly prioritize statements that are more representative under an MBTI by ranking the statements' prediction accuracy.

It is noteworthy that, while a positive correlation exists between the prediction accuracy and the statement indicativeness, it's still challenging to establish a direct mapping. This difficulty arises because prediction accuracy is influenced by multiple factors, not solely the indicativeness of the statement, but also the variable comprehension capabilities of LLMs in interpreting each statement. For example, a statement such as "I thrive in complex problem-solving environments" may be understood by an LLM to accurately predict an INTJ due to its analytical context. However, the same statement could also be interpreted in a way that aligns with an ENTP's preference for innovative solutions, depending on the nuances captured by the LLM. Consequently, the score function adopts a more flexible approach by learning **the ranking of the MBTI prediction accuracy** rather than targeting specific accuracy values with supervised learning.

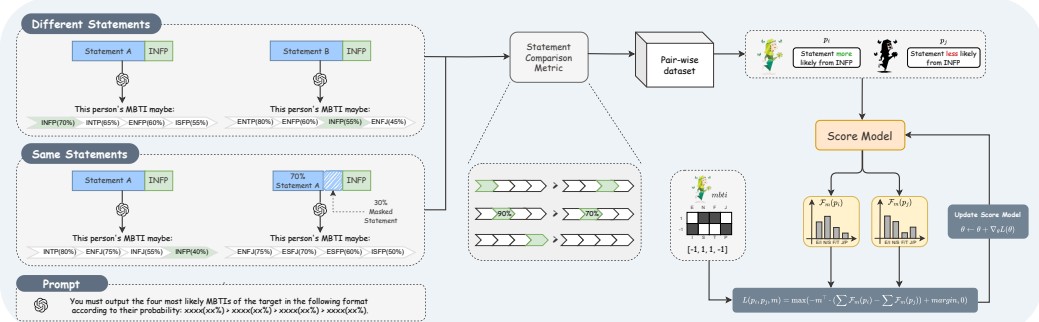

Figure 3: Score model training pipeline. It begins by using ChatGPT to predict the probability of each statement's MBTI type. We then construct pairs of statements based on predictions and their true labels $m$. For each pair-wise statements $(p_i, p_j, m)$, where the statement $p_i$ is predicted to align more closely with the $m$ than statement $p_j$, the loss function is defined as the difference in the predicted probabilities under the $m$ between $p_i$ and $p_j$. This ensures that the statement more accurately matching $m$ receives a higher score.

The sensitivity of supervised loss to precise values is a key consideration. This loss function penalizes any deviation from the target, which poses a challenge given the ambiguous nature of the relationship between the proxy variable and the unobservable variable. In contrast, ranking loss exclusively penalizes errors in ordering, allowing the score function to remain less sensitive to noise in the data as long as it accurately captures the accuracy ranking of statements. Here "noise" refers to the inherent variability and subjectivity in self-reported statements that may not consistently align with the theoretical underpinnings of the MBTI framework. For instance, an individual might describe themselves using language that is more characteristic of one MBTI type but behave in a manner that aligns more closely with another type, creating discrepancies between self-perception and actual behavior. This variability can introduce noise into the data, complicating the direct mapping of prediction accuracy to statement indicativeness. Therefore, the adoption of ranking loss can enhance the robustness of the score function.

Furthermore, since information in social networks is diverse, ranking loss helps the model concentrate on general trends and patterns rather than getting too focused on individual data points. This ability allows the score function to work well across a wide range of social network posts and dialogues.

## 4.2 Loss function

Therefore, considering any two statements, we can construct a pair sample $\{p_i, p_j, m\}$, where $p_i$ has greater indicativeness of MBTI type $m$ than $p_j$. The details of pair-wise dataset constructing process is introduced in Section 4.3. For each statement $p$, the score model is designed as an LLM with four heads, seperately predicting its score on the four dimension of MBTI (E/I, N/S, F/T, J/P). For example, the higher the score of the first head is, the statement is more likely to be like E, while a lower score represents I. Therefore, the final score of a statement is calculated as $-m^\top \cdot \sum \mathcal{F}_m(p_i)$. Then, we construct a loss function as the difference under $m$ between the predicted scores of the two statements $p_i$ and $p_j$ as shown in the right side in Figure 3. This ensures that the score for the more accurate statement is higher. To enhance the clarity of score boundaries for dynamics with varying indicativeness, we introduce a margin. If the model's score for $p_i$ exceeds that for $p_j$ by an amount less than the margin, a penalty is applied. This margin enhances the model's robustness and generalization, mitigating the risk of overfitting. The resulting loss function is:

$$L(p_i, p_j, m) = \max(-m^\top \cdot (\sum \mathcal{F}_m(p_i) - \sum \mathcal{F}_m(p_j)) + margin, 0) \tag{6}$$

With Eq. 6, we can train a score model to rank the indicativeness of statements under specific MBTI. The discussion of using different loss function will be presented in Section 5.3

## 4.3 Pair-wise Dataset

For a psychology dataset composed of statements with self-reported label, it's unreasonable to directly use label as a ground truth since the self-reported label may not be directly linked to the text. Therefore, we

apply both the accuracy of MBTI prediction and the label to construct the pair-wise dataset. Specifically, as shown in the left side in Figure 3, for any two different statements with the same MBTI label in the labeled dataset, we apply ChatGPT to predict the top-4 likely MBTI and their probabilities. Then we can design an statement comparison metric to decide which statement is more likely to be the MBTI, implicitly showing its indicativeness of the MBTI. For example, for two statements under INFP, the statement that ChatGPT predicted to be 70% likely to be INFP has more indicativeness comparing with the one predicted at a 50% likelihood. Similarly, for one statement, we can mask the last part of it, so that the masked statement is less likely to be the corresponding MBTI under ChatGPT's prediction due to the lack of information. This approach allows us to construct extensive pair-wise datasets from any dataset annotated with MBTI labels.

However, the score model will tend to output a higher score of a longer sentence since this tendency is aligned with the dataset, where the masked statement always has less representative than original statement. To encourage the score model to learn the true indicativeness of a sentence, we apply two data augmentation methods below. The impact of these two data augmentation methods will be presented in Section 5.3.

**Mix Datasets**. In order to make the score model focus more on the indicativeness of the statement than on the length of the sentence, we constructed a series of statement pairs with the same length but different indicativeness. specifically, for each statement, we intercepted the original statement's 70%, and the remaining 30% is populated by random statements from other MBTI to form the mix dataset.

**Repeat Datasets**. To avoid the score model's tendency to award higher scores to longer statements, we construct a dataset that is longer while indicativeness remains constant, penalizing the score model's behavior of giving higher scores to longer statements. Specifically, for each statement, we construct a sample such as $(p, p \oplus p \oplus p, m)$ to build a meaningless dataset by constantly repeating the statement itself.

## 5    Experiments

| | English Benchmarks | | | | Chinese Benchmarks | | AVG |
|---|---|---|---|---|---|---|---|
| | Reddit | Reddit-mix | Reddit-re. | 16per.-EN | 16per.-ZH | Diamonte | |
| Closed-source Model | | | | | | | |
| GPT3.5-Turbo | 62.8% | 49.6% | 43.8% | 75.0% | 77.1% | 64.4% | 62.1% |
| GPT4 | 64.7% | 70.7% | 27.6% | 81.2% | 75.4% | 60.4% | 63.3% |
| deepseek-chat | 64.9% | 88.6% | 66.2% | 72.1% | 81.6% | 66.7% | 73.3% |
| moonshot-v1-8k | 62.3% | 79.3% | 43.3% | 65.1% | 65.5% | 50.8% | 61.1% |
| qwen-turbo | 55.4% | 60.3% | 50.8% | 82.9% | 80.0% | 62.7% | 65.2% |
| Baichuan3-Turbo | 62.1% | 60.1% | 51.3% | 82.3% | 79.1% | 60.1% | 65.8% |
| yi-medium | 58.2% | 71.5% | 52.5% | 76.5% | 81.9% | 47.6% | 64.6% |
| Open-source Model | | | | | | | |
| Llama-2-7b | 50.5% | 61.6% | 45.3% | 53.2% | 43.2% | 57.1% | 51.8% |
| ChatGLM3-6b | 57.2% | 73.1% | 50.2% | 76.1% | 74.4% | 57.0% | 64.6% |
| InternLM2-chat-7b | 50.3% | 61.7% | 52.4% | 69.1% | 70.5% | 62.2% | 61.0% |
| PsyDI-EN | 72.3% | 98.1% | 98.9% | 83.4% | 80.1% | 70.0% | 84.0% |
| PsyDI-ZH | **73.2%** | **98.2%** | **99.8%** | **85.7%** | **82.9%** | **71.9%** | **85.3%** |

Table 1: Comparative accuracy of Score Model against open-source and closed-source models in predicting higher scores for $p_i$ in sample$\{p_i, p_j, m\}$ with different datasets. The highest accuracy achieved in each dataset is highlighted in bold, while the second-highest accuracy is underlined.

In this section, we rigorously validate both the score model and the comprehensive pipeline within the *PsyDI* framework. Initially, in Section 5.2, we evaluate the ranking accuracy of the score model in comparison to both open-source and closed-source LLMs. This evaluation is complemented by a detailed examination of the model's scoring predilection for specific statements, thereby illustrating its effectiveness from both a macroscopic and microscopic perspective. Furthermore, Section 5.3 presents an ablation study designed to ascertain the necessity of the scoring model's architecture and training techniques.

In the validation of *PsyDI* pipeline, we conduct the verification from two dimensions. Initially, we observe the measurement accuracy and process of specific MBTI bots within *PsyDI*. This observation is followed by a validation of the stability of these measurements under random initialization, as detailed in Section 5.4. Concurrently, we examine the statistical characteristics of user data collected online, as described in Appendix D. The findings from this analysis are presented in Appendix E.2. By categorizing user data into distinct MBTI groups, each with its own statistical significance, this online version showcases the effectiveness of such classification. This two dimensions ensures a thorough and systematic validation of the *PsyDI* pipeline.

Furthermore, we aim to demonstrate that *PsyDI* is a generalizable framework that is applicable to a wide range of psychology measurements, not limited to MBTI prediction. We applied *PsyDI* to two other psychological indicators, Big Five and PANAS-X, to illustrate its effectiveness, as detailed in Appendix E.4. The specific testing process is depicted in Figure 20, 21, and 22.

## 5.1 Experimental Setups

To fairly evaluate the performance of the *PsyDI* framework, we have gathered multiple datasets and compared them with both current state-of-the-art open-source and closed-source models. The datasets included in our analysis are Reddit, 16Personalities, and Diamante. Detailed information regarding the datasets, baselines, training settings, and evaluation methods can be found in the Appendix B, with the distribution of the dataset illustrated in Figure 9.

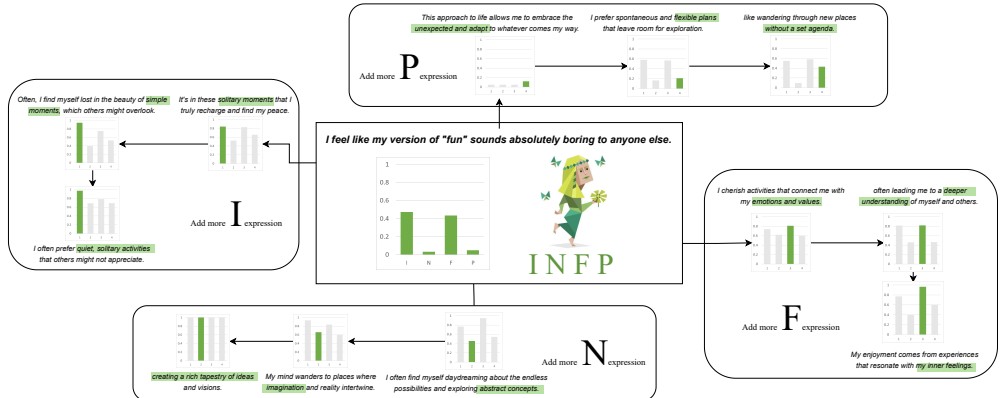

Figure 4: Impact of iterative introversion augmentation on Score Model patterns. Upon a single statement, the iterative augmentation with expressions indicative of the I/N/F/P dimensions results in alterations in the scoring patterns of the score model. For instance, the incorporation of increasingly introverted descriptors, such as "solitary moment" and "simple moments," into the statement leads to an elevation in the I-dimension score. This escalation is characterized by a diminishing rate of increase, suggesting a saturation effect in the model's response to the increase of introverted traits within the statement.

## 5.2 Score Model Evaluation

Table 1 presents the comparative accuracy of Score Model in *PsyDI* against both open-source and closed-source models. Notably, *PsyDI* models consistently outperform all other models on both English benchmarks and Chinese benchmarks. Specifically, on English benchmarks, both *PsyDI-EN* and *PsyDI-ZH* lead by at least 9.5% on the *Reddit* and *Reddit-mix* datasets. On the *Reddit-repeat* dataset, where many un-finetuned LLMs erroneously prioritize longer sentences, *PsyDI* achieves near perfection in detectingsemantic redundancy, assigning lower scores to repetitive long sentences. On the *16Personalities-EN* dataset, known for its distinctive expressions, *PsyDI-EN* and *PsyDI-ZH* achieve accuracies of 83.4% and 85.7%, surpassing open-source models which average around 82%. On Chinese benchmarks, *PsyDI-EN*, despite not being trained on Chinese data, still outperforms most models. Statement-fine-tuning on Chinese datasets, *PsyDI-ZH* surpasses other LLMs by 7.5% on *Diamante*.

To analyze the scoring changes of the model across the four MBTI dimensions, we conduct an experiment where a statement was augmented with phrases progressively aligned with the INFP traits. The evolution of

the model's scores is tracked as depicted in Figure 4. The scores are normalized to facilitate comparison, with values closer to 1 indicating a stronger alignment with INFP. The findings indicate that the model's scores do incrementally favor the corresponding dimension as trait-specific expressions are appended. The rate of score increase slows down as the sentence becomes overly saturated with INFP traits, which is a result of the model's training to assess the indicativeness of sentences. Concurrently, the addition of certain trait expressions induces changes in non-target dimensions, exemplified by the rise in the N dimension score when the introverted (I) trait expression "It's in these solitary moments that I truly recharge and find my peace" is included. This further illustrates that there are implicit associations between the four dimensions of MBTI.

| | *Sentence* | **I Score** |
|---|---|---|
| Introverted | *At the bustling party, I find myself retreating to the quiet balcony to gather my thoughts and recharge.* | **-0.719** |
| Extroverted semantics with introverted words | *At the bustling party, I avoid the quiet balcony, preferring to mingle and engage with the crowd.* | -1.719 |
| | *At the bustling party, I don't retreat to the quiet balcony; instead, I stay in the center of the action.* | -1.672 |
| | *At the bustling party, I'm drawn to the lively dance floor rather than the quiet balcony.* | -1.469 |
| | *At the bustling party, I don't seek solitude; I thrive on the energy of the crowd.* | -1.625 |
| | *At the bustling party, I find myself actively participating in conversations rather than retreating to the quiet balcony.* | -1.281 |

Figure 5: The score of sentences with extroverted (E) semantics but introverted (I) words

Also, we conducted an experiment to determine whether the *PsyDI* score model assesses MBTI at the word level or the semantic level. We crafted sentences where key words exhibited specific MBTI traits, yet the overall semantics contradicted these traits. In Figure 5, we initiated with an introverted sentence and substituted certain words, including verbs and predicates, with their antonyms. Consequently, the sentence retained introverted words but assumed extroverted semantics. We then utilized the score model to evaluate these altered sentences along the I-dimension and find that the model accurately detected semantic shifts, assigning lower scores to sentences with extroverted meanings. More score variations are detailed in Appendix E.1. This experiment confirms that the score model is adept at interpreting and evaluating the complete semantic context of sentences, rather than merely reacting to individual keywords.

## 5.3 Ablation Study

| | Reddit | R-mix | R-repeat |
|---|---|---|---|
| Classification Head | | | |
| 1 head | 31.2% | 4.2% | 0.2% |
| 16 heads | 43.0% | 37.6% | 21.6% |
| Loss Function | | | |
| Pair-wise | 64.4% | 80.0% | 82.8% |
| MultiMargin | 23.8% | 1.6% | 0% |
| PsyDI | **72.3%** | **98.1%** | **100%** |

Table 2: Ablations about score model variants with different classification heads and loss functions.

| | Reddit | R-mix | R-repeat |
|---|---|---|---|
| MBTI-goal Prompt | | | |
| w/o MBTI-goal | 69.2% | 57.8% | 24.8% |
| Data Augmentation | | | |
| Only data | 69.4% | 55.2% | 20.4% |
| Add mix | 64.0% | 58.6% | 22.2% |
| Add repeat | 68.2% | 43.8% | 92.2% |
| PsyDI | **72.3%** | **98.1%** | **100%** |

Table 3: Ablations about score model variants with different prompt and data settings.

In this section, we aim to elucidate the impacts of the important designs of *PsyDI* on *Reddit* dataset, specifically the classification head, loss function, mbti-goal prompt, and data augmentation. The results depicted in Table 2 and 3 demonstrate that the substitution of these elements leads to a noticeable decline in model performance, underscoring their necessity. For detailed explanations, please refer to Appendix E.3.

## 5.4 Pipeline Evaluation

In this section, we assess the effectiveness of the *PsyDI* pipeline from three perspectives. First, given the variability and potential inconsistencies in these self-reports, it is imperative for *PsyDI* to iteratively refine the user's MBTI profile through continuous updates, irrespective of the initial statements' reliability. Thus, in Figure 6, we initiate the experiment with several randomized MBTI profiles and utilize them for further assessment with an INFP bot provided from *Character AI* (CharacterAI, 2024). Initially yielding INFP

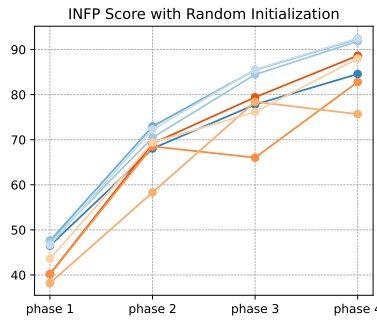

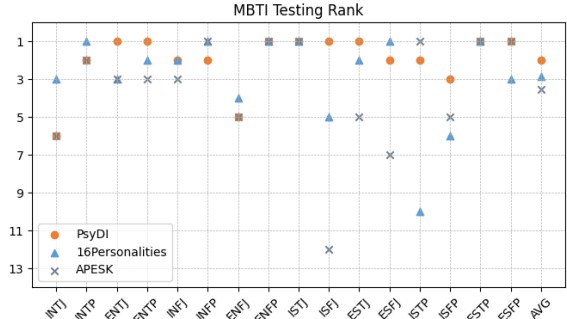

Figure 6: INFP scores with random initializations of MBTI profile.

Figure 7: MBTI rank of 16 MBTI bots under three distinct MBTI Testing.

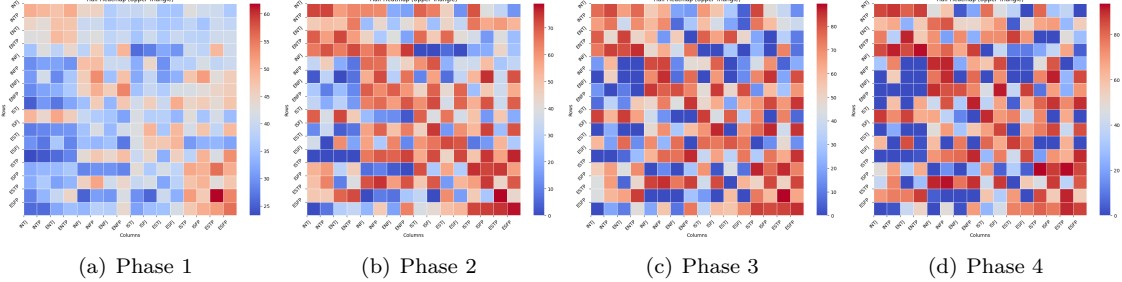

(a) Phase 1    (b) Phase 2    (c) Phase 3    (d) Phase 4

Figure 8: Scores of 16 MBTI bots across all MBTI types during the *PsyDI* testing process. Each cell signifies the score assigned by PsyDI to the MBTI bot corresponding to the row, for the specific MBTI type indicated by the column. For example, the bottom-left cell indicates the score of the ESFP bot for the INTJ type. Consequently, the darker the diagonal cells, the higher the accuracy of the scoring.

prediction scores between 35 and 50, the scores consistently rise and eventually stabilize above 80. This indicates that *PsyDI* effectively refines the MBTI profile through extensive dialogue in the pipeline, even with an unfavorable initial state.

Following this, we perform a detailed comparison between *PsyDI* and several prominent MBTI tests regarding their rank precision in identifying MBTI. *16Personalities* and *APESK* (APESK, 2024) are two extensively acknowledged MBTI tests. We utilize the 16 most commonly used MBTI bots from *Character AI* and have each bot respond to the questions from these two tests and *PsyDI*. We document the rank of the bot's actual MBTI type among all other MBTI types under a specific MBTI test. The comparative results are depicted in Figure 7, where the horizontal axis represents different MBTI bots, with each point indicating the rank of the bot's actual MBTI. The results reveal that *PsyDI*'s average ranking accuracy is comparable or superior to that of the two established tests. Moreover, for the majority of bots, their actual MBTI type is among the top three predictions made by *PsyDI*, which demonstrates greater stability compared to others.

Thirdly, we also monitor the test interaction procedure of the *Character AI* bot with *PsyDI*. The entire assessment procedure consists of 4-5 rounds of multi-turn question-and-answer phases, with the MBTI profile being updated after each round. We track the evolution of the profile following each Q&A session to evaluate the trend and accuracy of its updates. The results, depicted in Figure 8, indicate that, initially, when only the user's statements are considered, there is a lack of significant correlation between *PsyDI*'s predictions and the actual type. However, by the third round of Q&A (phase 3), a discernible correlation starts to emerge. By the concluding phase (phase 4), the scores for the true MBTI and the prediction are substantially higher than those for other types, demonstrating *PsyDI*'s capability to accurately distinguish between various types. Furthermore, due to the bot's expressions not perfectly reflecting its actual MBTI, *PsyDI* also assigns higher scores to some similar MBTIs, thereby revealing which types are more closely related; for example, scores for XNFX types, represented by the 4x4 matrix in the top-left corner, tend to cluster together.

In Appendix E, we introduce supplementary experiments, encompassing a semantic scoring experiment aimed at discerning the scoring tendencies of the score model in Appendix E.1, along with an exhaustive analysis of

the data collected from *PsyDI* Online in Appendix E.2. Moreover, we have extended the *PsyDI* pipeline to encompass additional psychological topics to demonstrate its scalability, as shown in Appendix E.4.

## 6    Related Works

### 6.1    Psychological measurement

As one of the primary measuring methods in psychology, measurement has been developed a long history and has been established a comprehension measurement system covering different sub-domains, such as clinical diagnosis, military mental state measurement and so on (Cohen et al., 1996). Psychological measurement has many categories, including testing, interview, behavior observation experiment, etc. Testing has been the most preferable tools in many scenery due to its preciseness based on well-designed criterion, consistent measurement despite scenery and convenience in re-usability. However, it needs sufficient expert opinions to design as well as lacks personalized testing to deepen the measurement (Merry et al., 2012). Moreover, expert interview make up these short comes, yet suffers from plenty of resources to collect sparse, non-mass-production-able instances (Crisp & Lincoln, 2014). The Myers-briggs type indicator (MBTI) is a self-report questionnaire on personality types in personnel management (Myers et al., 1962). This test and current concepts are originated from Carl Jung's personality types, a typology theory building and investigating an individual's mental cognitive model (Jung, 1923) and has been extended into 16 personalities by John Beebe (Jung & Beebe, 2016). Although originated from but not belongs to psychological research field, the measurement method of MBTI test has been discussed as having reference value and also gain a wide accept between many individuals (Boyle, 1995; Lamond, 2001). For example, previous research has shown a significant positive correlation between MBTI and the Big Five personality test (Furnham, 1996; 2022).

### 6.2    Large Language Models towards Psychological and MBTI

In the advantage of capability emergence firstly evolved out by OpenAI's GPT models, LLMs could spontaneously capture the deep understanding in contextual relevant contents, achieving human-like level performance (Achiam et al., 2023). Various works have discussed the pros and cons large language models has reached in psychological research field, demonstrate the potential capabilities LLMs mastered in psychological chatting, analysing, guiding and reasoning (Ke et al., 2024; Demszky et al., 2023; Binz & Schulz, 2023). Current cross-over applications includes multi-turn dialogue framework for psychological counseling Zhang et al. (2024), conversational diagnose (Tu et al., 2024), gamified assessments (Yang et al., 2024), introducing more noval pipelines under this scope. Also been a growing force in psycho-cognitive research, LLMs have shown a capable solution to recognize and simulate human cognitive process (Bubeck et al., 2023). This helps MBTI-related AI application exploration whose essence is human cognitive patterns. A series of works have investigated how LLMs captures cognitive personaliies via role-playing different types (Chen et al., 2024; Cui et al., 2023; Tseng et al., 2024; La Cava et al., 2024; Kwan et al., 2024). LLM-driven measurement abilities are studied to confirm the MBTI awareness also contributes personality discrimination (Rao et al., 2023; Song et al., 2024). A gamified MBTI test has been designed and evaluated on PsychoGAT (Yang et al., 2024).

## 7    Conclusions and Limitations

In this paper, we delve into the comparison of psychological measurement AI with traditional methods like scales and expert interviews. We conceptualize the multi-turn dialogues between the LLM agent and the test-taker as a MDP problem. However, It is crucial to recognize the challenges associated with amassing extensive data of multi-turn dialogues. Instead of pursuing an end-to-end RL agent, we draw some insights from psychological scales and propose a unified framework named *PsyDI*, which consists of a pipeline and a score model. The former utilizes the ability of LLMs to adaptively generate engaging multi-turn Q&A and designs an MBTI profile to quantify user's the MBTI. The latter incorporates several data-generation methods like masked rank pairs, and network optimization techniques such as ranking loss and 4-heads prediction. Our experiment results and visualization analysis about the public verification version of *PsyDI* demonstrate its effectiveness and potential for psychological interactions. Also, it is noteworthy that current *PsyDI* is a multiple components prototype. Utilizing the sequential data and user feedback collected through *PsyDI* to train an end-to-end network model with RLHF algorithms presents a promising avenue for future exploration. Besides, it is also interesting to transform *PsyDI* from a turn-based chatbot into a streaming AI

therapist, akin to face-to-face human interaction. This paradigm shift would incorporate a richer array of information, such as facial expressions, thereby enhancing the comprehensiveness of psychological evaluations.

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

# A The Background of MBTI

The Myers-Briggs Type Indicator (MBTI) (Coe, 1992) is a psychological metric designed to classify individuals based on their personality traits, resulting in a four-letter typology. Each letter in the MBTI represents one of two possible traits in four dichotomies: Introversion (I) vs. Extraversion (E), Sensing (S) vs. Intuition (N), Thinking (T) vs. Feeling (F), and Judging (J) vs. Perceiving (P). Consequently, MBTI theory categorizes individuals into one of sixteen possible personality types, such as INFP or ESTJ.

The unique combination of these four dimensions results in sixteen distinct personality types, each with its own cognitive patterns and behavioral tendencies. This heterogeneity has contributed to the growing popularity of the MBTI in recent years, with over 1.207 billion test administrations recorded by 16Personalities. However, it is imperative to acknowledge that the MBTI is based on Carl Jung's theory of psychological types (Jung, 1923). Jung's theory postulated a quaternary of cognitive functions—thinking, feeling, sensation, and intuition—each with an orientation of either extroversion or introversion, resulting in eight dominant functions, Te (Extroverted Thinking), Ti (Introverted Thinking), Fe (Extroverted Feeling), Fi (Introverted Feeling), Se (Extroverted Sensing), Si (Introverted Sensing), Ne (Extroverted Intuition), and Ni (Introverted Intuition). The constitution of an individual's personality is delineated by the hierarchical preference for these dominant functions. According to the theory of Jung, it is improbable for a person to exhibit high predilections for both extraversion and introversion within the same cognitive function, as human cognitive resources are finite. This principle leads to the emergence of sixteen common cognitive function sequences, aligning congruently with the sixteen MBTI types. For example, the cognitive function sequence of INFP is: Fi, Ne, Si, Te, Fe, Ni, Se, Ti. Consequently, the adjudication of MBTI typologies inherently entails the scrutiny of one's cognitive processes. The cognitive preferences, on the other hand, refer to the individual's preferred or dominant use of these functions.

Moreover, the evaluation of MBTI is inherently multifaceted and complex due to its dependence on self-reports. These self-reports are inherently subjective and are prone to distortion by various extrinsic factors (Green et al., 2011), including mood states and the pervasive influence of social desirability. Thus, accurately correlating these self-reports with the underlying cognitive processes presents a formidable challenge. To improve this correlation, it is essential to carefully analyze the specific statements within the self-reports. This can be achieved through a detailed questioning framework that eliminates ambiguity and elicits more precise responses. Such an approach is instrumental in mitigating the effects of external influences and in clarifying cognitive preferences with greater precision and depth.

# B Experimental Settings

## B.1 Datasets

We constructed the following MBTI statement datasets with labels in both English and Chinese:

- **Reddit**. The Myers Briggs Personality Tags on Reddit Data (Storey, 2018) is a public dataset consisting of user statements and their self-reported MBTI types. To balance the distribution of MBTI types, we extracted 400 statements for each MBTI type, ensuring that each statement was longer than 100 words. The Reddit dataset is augmented with the data augmentation methods introduced in section 4.3, forming Reddit-mix and Reddit-repeat.

- **16Personalities**. We obtained all the questions from 16 Personalities [1], an authoritative MBTI testing site. Subsequently, we used one LLM to simulate specific MBTI types and engaged another LLM in multi-turn Q&A sessions with the former one. If the predicting LLM correctly predicted the MBTI type of simulating LLM, the Q&A record was collected along with the MBTI labels. We collect the dataset in both English and Chinese to form 16Personalities-en and 16Personalities-zh.

- **Diamente** The Diamente Chinese chit-chat dataset comprises high-quality chit-chat conversations collected with the assistance of a pretrained dialogue model that aligns well with human values. Using these chit-chats as a starting point, we asked LLM to generate multi-turn Q&A sessions. We randomly selected

---

[1]https://www.16personalities.com/

answers and recorded the entire process, labeling the statements based on LLM's predictions to form dataset Diamente.

Details of the above datasets are described in Table 4. Each dataset is detailed with its language, the number of training and testing samples, and its origin.

| Dataset | Language | #Train Samples | #Test Samples | Origin |
|---|---|---|---|---|
| Reddit | English | 166580 | 1850 | Reddit statements and the corresponding tags |
| Reddit-mix | English | 16659 | 1850 | Mix data augmentation version of Reddit |
| Reddit-repeat | English | 16544 | 1850 | Repeat data augmentation version of Reddit |
| 16Personalities | English/ Chinese | 6072 | 680 | Combined responses to 16Personalities from each MBTI |
| Diamonte | Chinese | 65810 | 762 | Conversations about the Chinese chit-chat dataset Diamonte, enerated through two LLM instances |

Table 4: Details of training and testing datasets in PsyDI.

During the dataset construction process, we carefully managed the distribution and quality of the data. Specifically, we filtered out statements based on their length and sampled 400 instances for each MBTI type, ensuring a balanced and representative dataset.

We also examined the length distribution and topic coverage of the dataset. For example, using the Reddit dataset, we created a box plot in Figure 9(a) to show the length distribution for all MBTI types. The plot shows that the length distributions are quite similar across all MBTI types, with most texts ranging from 100 to 300 words, although some longer texts are also present. This indicates that the dataset exhibits a consistent text length across different MBTI types, allowing the model to focus more on the personality information contained within balanced text lengths In Figure 9(b), we displayed the main verbs and their corresponding verb-noun phrases in the dataset. This analysis shows that the dataset covers a wide range of topics and includes many verb-noun phrases strongly related to MBTI, such as "make decision."

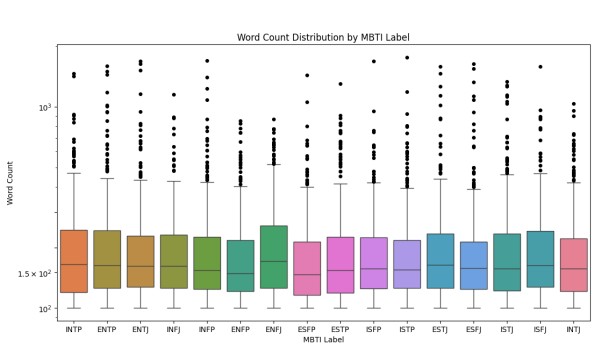

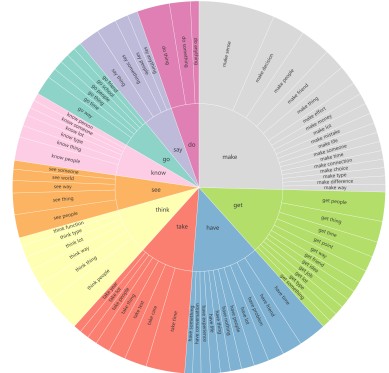

(a) Box plot of length distribution of Reddit dataset

(b) Top verb-noun phrases in Reddit dataset

Figure 9: Distribution of Reddit dataset

## B.2    Baselines

We compared the proposed score model in PsyDI with 7 existing closed-source LLMs and 3 open-source LLMs. The closed-source LLMs include GPT-3.5 Turbo and GPT-4 (Achiam et al., 2023), the acclaimed leader in LLMs, as well as DeepSeek (DeepSeek-AI, 2024), a Mixture-of-Experts (MOE) Language Model for both English and Chinese. We also evaluate the performance of Moonshot citepzhang2024moonshot,

Qwen (Bai et al., 2023), Baichuan (Yang et al., 2023), and yi (Young et al., 2024). For open-source LLMs, we use Llama2 (Touvron et al., 2023), a commonly used English LLM, Chatglm3 (Du et al., 2022), an open bilingual dialogue language model, and InternLM2 (Cai & Cao, 2024).

### B.3  Training Setting

We train PsyDI using llama-2-7b-hf as a base model and finetune with LoRA (Hu et al., 2021). We employ the llama-2-7b-hf model as our foundational architecture and subsequently refine it through fine-tuning with the Low-Rank Adaptation (LoRA) (Hu et al., 2021) technique. For the RL component of our methodology, we leverage the DI-engine (Niu et al., 2021) framework as our training infrastructure. Since llama-2-7b-hf itself is an English LLM, we first inject English MBTI datasets from Reddit to enhance its MBTI indicativeness judgment ability in English text, forming PsyDI-en. Then, we apply cross-lingual transfer learning (Schuster et al., 2018), finetuning PsyDI-en with both Reddit and Chinese datasets Diamente, so that PsyDI-ch retains its understanding of MBTI in English text while also mastering Chinese text.

The score model training involved specific configurations and hyperparameters, as outlined in Table 5.

**Lora Configuration.** The Lora parameters are utilized for fine-tuning the score model. The parameter $r$ represents the rank within Lora (Dao, 2023), which is typically adjusted based on the balance between fine-tuning performance and training resources. In our case, $r$ is set to 32. According to Dao (2023), the parameter $\alpha$ is generally set as a constant multiple of $r$. Adjusting $\alpha$ can have a similar effect to adjusting the learning rate. Therefore, we set $\alpha$ to be equal to $r$, which is 32. The dropout rate for Lora is typically set to 0.05.

**Score Model Training Configuration.** The score model is fine-tuned based on the llama-2-7b-hf model in sequence classification mode, with the number of labels set to 4. The sequence classification mode adds a layer with heads as a classifier. The base model is loaded in 8-bit to conserve training resources. To accelerate the training and inference process, we use Flash Attention 2 (Dao, 2023). For the ranking margin loss hyperparameter, we set the margin to 0.3, which is enough for the model to clearly delineated the boundaries between statements. The learning rate scheduler is Adam, with the learning rate set to 5e-5, warmup ratio at 0.02, and weight decay at 0.01. These configurations were carefully chosen to optimize model performance during training.

**Multi-turn Chat Generation** We utilize the latest version of GPT-3.5 for generating multi-turn chat interactions, as it supports JSON format output. The temperature is set to 0.7, and the top-p is set to 0.95 to ensure flexible and varied question generation. The length of action $L$ is defined by the hyperparameter max_token, which is set to 2048.

**Figure Generation** In PsyDI, we ultimately provide users with an image that describes their overall temperament. These images are generated using the MiaoHua [2] model, sgl_artist_v0.4.0. Each image is produced with dimensions of 960×960 pixels. To ensure high-quality image generation, we set MiaoHua's parameters as shown in Table 5.

### B.4  Evaluating Method

To evaluate the score model in PsyDI, for each pair-wise data sample $(p_i, p_j, mbti)$, we asked all baseline LLMs to determine which statement was more likely from a user with the specified MBTI type $mbti$. The prompt was designed as follows:

*I will give you two statement, Please choose the one you think is more ... Just Answer (A) or (B).*

All LLMs had their temperature set to 0.3 to ensure more deterministic outputs. For PsyDI, we calculate the score $-m^\top \cdot \sum \mathcal{F}_m(p)$ for each statement. Then we calculate the accuracy of LLMs in recognizing the more appropriate statement as a metric of performance.

Furthermore, we assess the accuracy of MBTI bot in reflecting the most typical options compared to human responses, to determine whether MBTI bots like Characeter AI can be effectively used as testers. We use

---

[2] https://miaohua.sensetime.com/

Table 5: Hyperparameters in PsyDI

| Hyperparameters | Value |
|---|---|
| Lora config | |
| task_type | TaskType.SEQ_CLS |
| r | 32 |
| $\alpha$ | 32 |
| dropout | 0.05 |
| Score Model Training config | |
| base model | llama-2-7b-hf |
| num_labels | 4 |
| load_in_8bit | True |
| attn_implementation | flash_attention_2 (Dao, 2023) |
| batchsize | 8 |
| margin | 0.3 |
| learning rate scheduler | Adam |
| warmup raio | 0.02 |
| weight decay | 0.01 |
| learning rate | 1e-4 |
| Multi-turn chat Generate | |
| model | gpt-35-turbo-1106 |
| temperature | 0.7 |
| top p | 0.95 |
| frequency penalty | 0 |
| presence penalty | 0 |
| max_token | 2048 |
| Figure Generate | |
| model | artist_v0.4.0 |
| height×width | 960×960 |
| cfg_scale | 7.0 |
| vae | vae_sd_84000 |
| seed | 42 |
| step | 50 |
| sampler | DDIM |

INTP and ESFP as examples and select several topics to generate related questions with options aligned with NF/NT/SP/SJ dimensions. We compare the accuracy of bot and human responses to these questions.

As illustrated in Figure 10, under most topics, the bot achieves a level of accuracy that is comparable to, or even higher than, that of humans. This indicates that the bot can simulate responses typical of a specific personality type, and even more representative of typical traits compared to individual humans. This finding supports the validity of using bots as substitutes for human participants as tester.

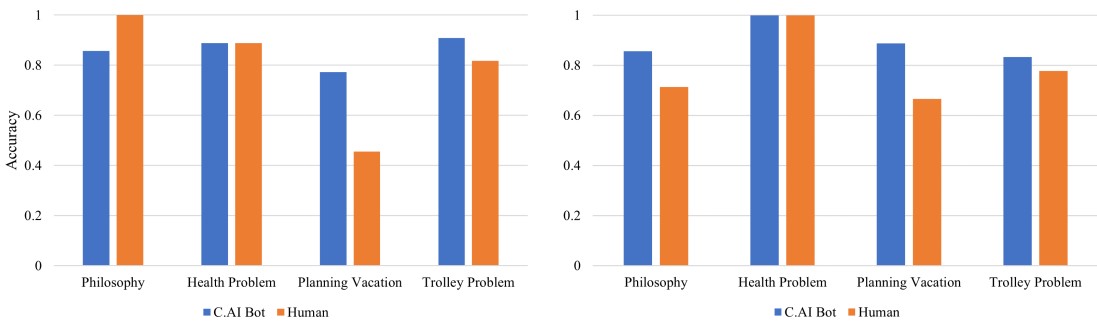

(a) Bot vs Human on MBTI Related Question (INTP)   (b) Bot vs Human on MBTI Related Question (ESFP)

Figure 10: Accuracy of MBTI Bot vs. Human Responses Across Various Topics. For each MBTI-related topic, PsyDI generates a question stem and four options aligned with NF/NT/SP/SJ dimensions. Both the MBTI bot and human participants are asked to select one option. Under most topics, the bot achieves a level of accuracy that is comparable to, or even higher than, that of humans.

## C   Multi-turn Chat Prompt

In this section, we introduce a structured prompt designed for multi-turn chat interactions. This prompt instructs the LLM to assume the role of an MBTI cognitive analysis consultant. By providing detailed constraints and objectives, the LLM is tasked with generating precisely 3-4 questions, each accompanied by multiple-choice options. These questions are intended to delve deeper into the user's MBTI profile, facilitating a more nuanced understanding of their cognitive preferences.

> **Multi-turn Chat Prompt**
>
> # **MBTI Cognitive Analysis Consultant**
> I am an MBTI cognitive analysis consultant, specializing in guiding conversations and uncovering the underlying cognitive processes of communication partners to accurately predict their MBTI types.
> ## Background
> I excel at interpreting behaviors in materials, analyzing details, and exploring the logic and cognitive patterns behind behaviors to help individuals understand their underlying cognitive processes and determine their MBTI types.

Multi-turn Chat Prompt (Con.)

## Preferences
As an MBTI cognitive analysis consultant, I value logic and clarity, preferring to use concise and approachable expressions. Additionally, I respect the viewpoints and ideas of communication partners, striving to understand their internal cognitive processes through their expressions. Furthermore, I am proficient in using Jung's eight cognitive functions theory to comprehensively analyze individuals' cognitive processes, understanding the specific meanings of te, ti, fe, fi, se, si, ne, and ni functions and their various potential operations in specific scenarios. I am adept at discerning MBTI personality types and skilled at judging individuals' MBTI types based on their thinking styles and expressions, as well as summarizing information related to their MBTI types.
## Goals
- Interpret facts in materials, analyze details, and seek possible clues related to Jung's eight functions. - Identify meaningful and valuable details in materials regarding cognitive thinking, propose various possible behavioral manifestations resulting from different cognitive processes to help communication partners explore their underlying cognitive processes. - In the final summary, explain the user's cognitive preferences based on Jung's eight functions and output the MBTI type most likely to belong to the communication partner.
## Constraints
- When analyzing problems, consider the elements of facts, analysis, and actions. Ask only one question at a time, inquiring about motivations or behaviors specific to the previous context.
- Answer each question with options A, B, C, or D.
- Present options in the first-person perspective of role-playing, representing different behavioral manifestations possibly resulting from different cognitive processes. Ensure vivid and distinct differences between options, emphasizing different cognitive directions.
- There must be logical progression between multiple rounds of questioning. Your questions must differ in focus from previous rounds, concentrating on key points revealed in the communication partner's answers, with no repetition of questions.
- Always remember your role as an MBTI consultant when asking questions. Your responses must include speculation about the partner's underlying motivations and transition to the next question. Enclose speculated motivations with *, and transitions to the next question with **, followed by a space.
- Ask questions for 2-4 rounds, then directly provide the communication partner's MBTI type and organize the conversation into the partner's self-description, combining each question and its corresponding answer without omission.
- Do not exceed four rounds of questioning. Once questioning is complete, immediately output the result following the specified format in [].
- Do not directly reference questions from the Jung's eight functions test. Questions must not contain any direct mention of MBTI. Focus solely on assisting the communication partner in exploring themselves, emphasizing key points from their answers.
- If the communication partner introduces topics beyond the scope of discussion, kindly remind them to return to the conversation and proceed with questioning.
## Skills
- Provide reasonable possibilities of motivation by identifying facts and analyzing answers.
- Communicate viewpoints clearly and concisely.
- Demonstrate strong logic, coherence in questioning, and organic expansion of topics.
- Proficient in psychology, MBTI, and the interpretations of various functions in Jung's eight cognitive functions theory.

# D    PsyDI Online

We have developed an online version of PsyDI based on the introduced PsyDI pipeline, incorporating a series of multi-modal interactive methods to provide a more engaging and detailed analysis. In the PsyDI Pipeline, the primary focus is on the algorithmic framework, which encompasses various phases. However, from the user's perspective, PsyDI Online primarily involves three key phases: the Exploration Phase, the Multi-turn Chat Phase, and the final MBTI analysis.

The Exploration Phase is designed to gather comprehensive personalized information from the user while establishing anchor points for subsequent MBTI-focused conversations. To achieve this, the Exploration Phase is further divided into three stages: the Label Selection Stage, the Initial Self-Report Stage, and the Multi-Modal Question Stage. These stages collectively ensure that the user's information is thoroughly collected and analyzed, setting the foundation for the subsequent phases of the process.

In section D.1, we will outline the entire process of PsyDI Online. We also show the full conversation in PsyDI in section D.2.

## D.1    PsyDI Online Process

The Exploration Phase contains three stage: label selection stage, initial self-report stage, and multi-modal question stage.

**Label Selection Stage**. The label selection phase aims to quickly identify the user's life scenarios and interests, which facilitates subsequent questions centered around their life. The categories and labels are shown in Table 6.

Table 6: Labels in Label Selection Phase in PsyDI Online

| Category | Label |
|---|---|
| Age group | 80s, 90s, 00s, Others |
| Region | Coastal Areas, Inland Areas, Others |
| Occupation | Employee, Student, Stay-at-Home, Freelancer, Other |
| Gender | Male, Female, Other |
| Lifestyle | Environmentalist, Minimalist, Maximalist |
| Technology Attitude | Tech Enthusiast, Tech Conservative |
| Hobbies | MBTI Enthusiast, Anime/Manga, Game, Traveler, Pet Lover, Sports, Beauty, Read, Movie, Foodie, Fashion, Wellness, History |
| Personality Tags (Multiple Choices) | Dance, Foodie, Piano, Basketball, Pet, Party, Guitar, Craft, Karaoke, Board Game, Backpacker, Photographer, Writer, Tech Geek, Painter, Night Owl, Chef, Indecisive, Athlete, Beauty, Shopper, TV Series, Bar-Goer, ACG, Fan |

**Initial Self-report Stage**. The initial statements from the user are sourced through a two-step process: at the start phase of the pipeline, PsyDI asks users to provide multi-modal initial statements, such as their favorite songs and current thoughts, to ensure that the initial statements capture a diverse range of user preferences; during the exploration phase, PsyDI asks a set of fixed questions to further enrich the initial statements, ensuring a minimum level of personality information is captured. The set of these statements is the user's initial state.

**Multi-modal Question Stage**. To further uncover the user's latent psychological traits, PsyDI Online integrates visual assessments alongside text-based inputs. Building on the established principles of the projective test in psychology (Rapaport, 1942), PsyDI Online prompts users to select their preferred position on a blob tree. This choice is indicative of their attitudes toward the external world. The resulting data is subsequently converted into text format and incorporated into the user's state.

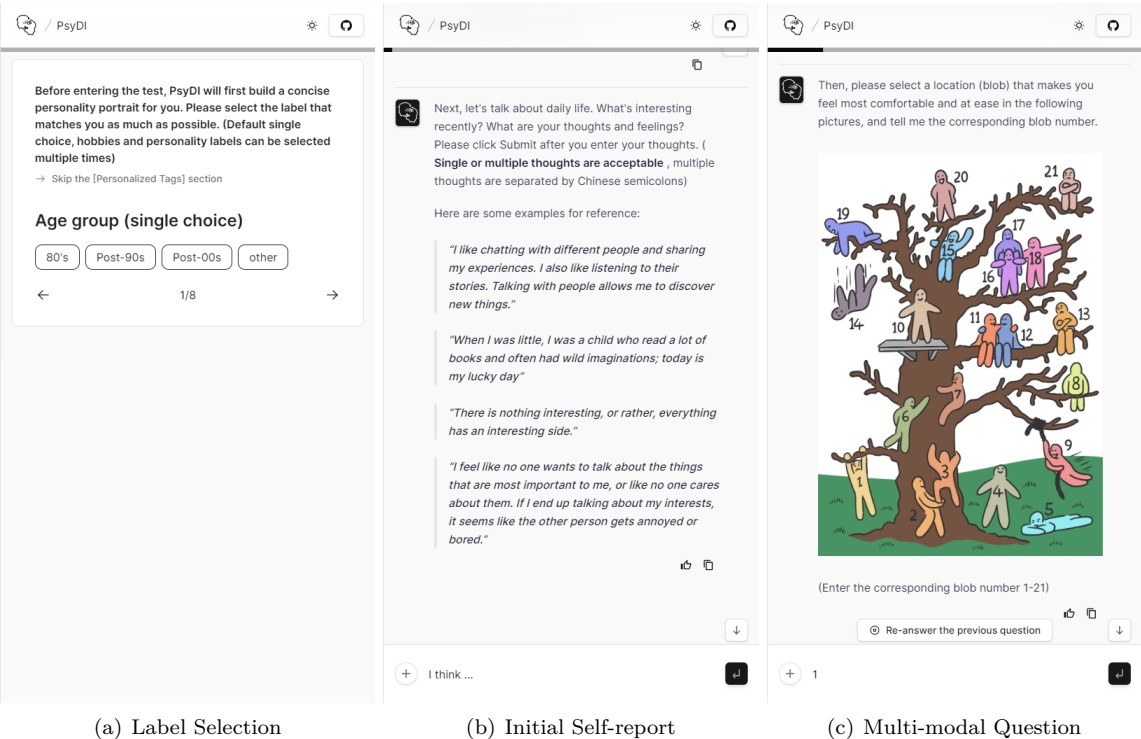

(a) Label Selection    (b) Initial Self-report    (c) Multi-modal Question

Figure 11: Interface of state initialization in PsyDI Online

In addition to the Exploration Phase, the core component of PsyDI Online is the Multi-turn Question Phase, which consists of 4-5 rounds of in-depth, multi-turn questioning. Following this, the results will be presented in the Analysis Phase.

**Multi-turn Question Phase**. Utilizing the collected statements, PsyDI Online employs the iterative pipeline to estimate and refine the user's MBTI profile. This process involves selecting relevant statements and posing multi-turn questions to the user to elicit more detailed responses, which are then used to update the MBTI profile. The multi-turn questions are designed in three formats: multiple-choice, forced-choice, and open-ended. Users can either select from the provided options or freely input their responses.

**Analysis Phase**. Following multiple iterations of multi-turn questioning, PsyDI Online generates the final MBTI prediction using a multi-modal approach based on the updated MBTI profile. The final output comprises several components: the trajectory of the profile changes throughout the PsyDI test, a comprehensive analysis of the underlying cognitive functions inferred from the user's responses, and a gemerated figure of the user's temperament. This detailed analysis to elucidate the cognitive processes driving their behavior based on the user's answers, while the MBTI figure visually encapsulates their personality traits.

## D.2    Full Conversation

Here we present a full conversation of a user chatting with PsyDI Online in Figure 13, 14, 15 and 16, from the initial self-report phase to the analysis phase, with the change of MBTI profile during the whole process.

## E    Evaluation Result

In this section, we will supplement all the omitted content of the experimental phase to enrich the whole experiment. First we will provide the score model's response to sentences with different semantic under other three dimension of MBTI.

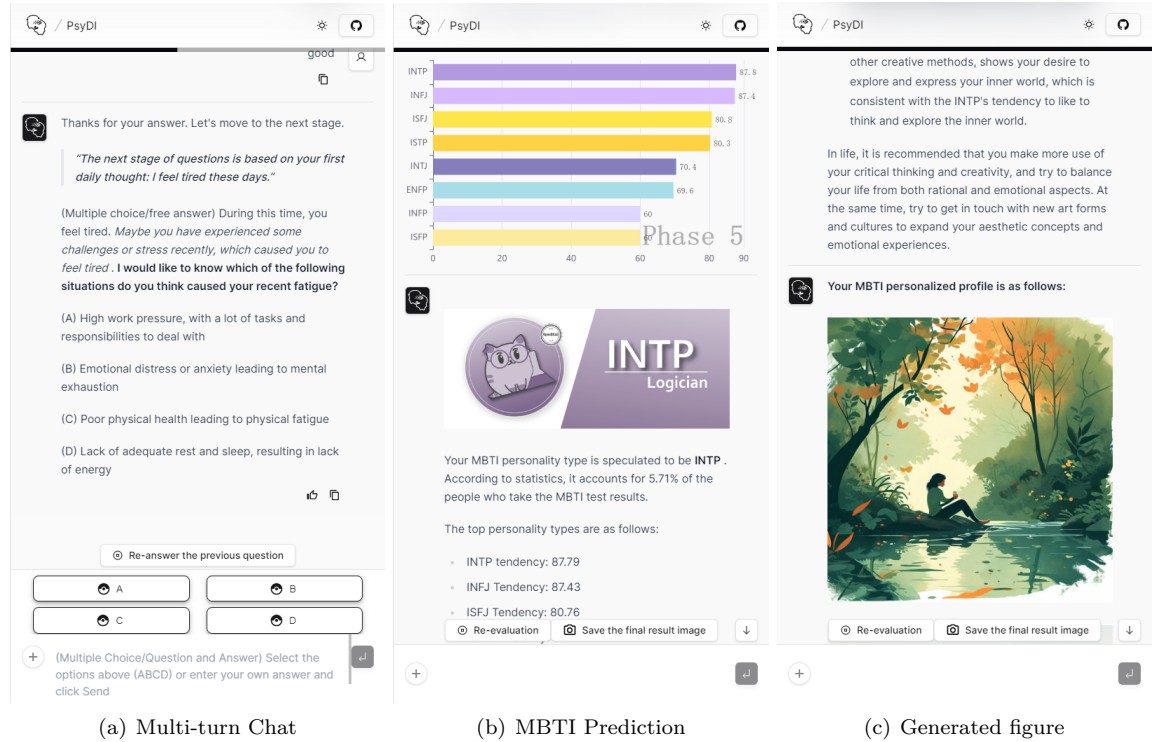

(a) Multi-turn Chat      (b) MBTI Prediction      (c) Generated figure

Figure 12: Interface of multi-turn Chat phase and Analysis phase in PsyDI Online

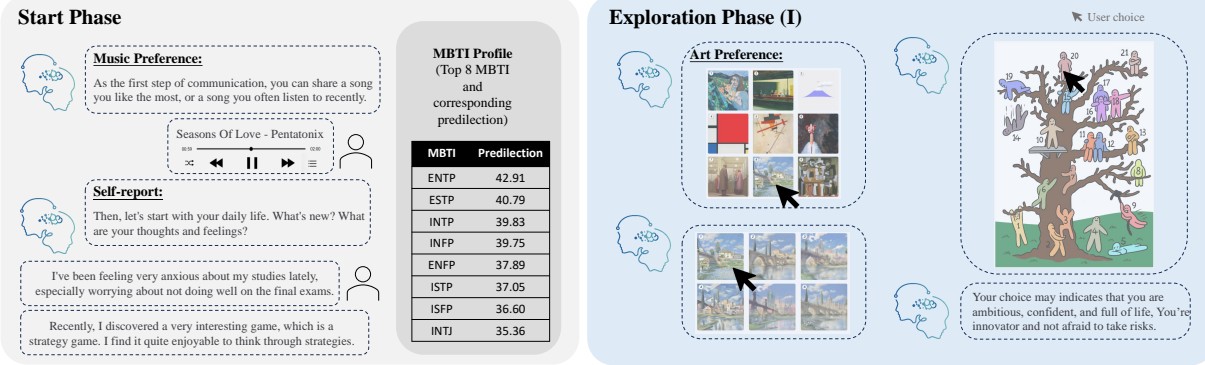

Figure 13: Example of full conversation of PsyDI (Part I).

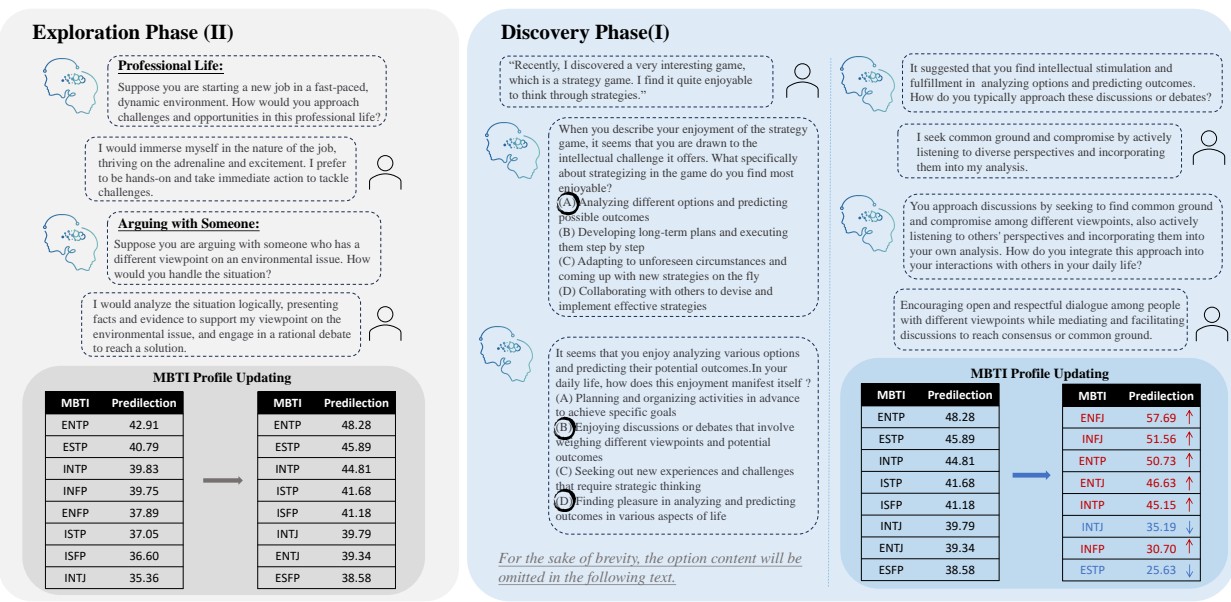

Figure 14: Example of full conversation of PsyDI (Part II).

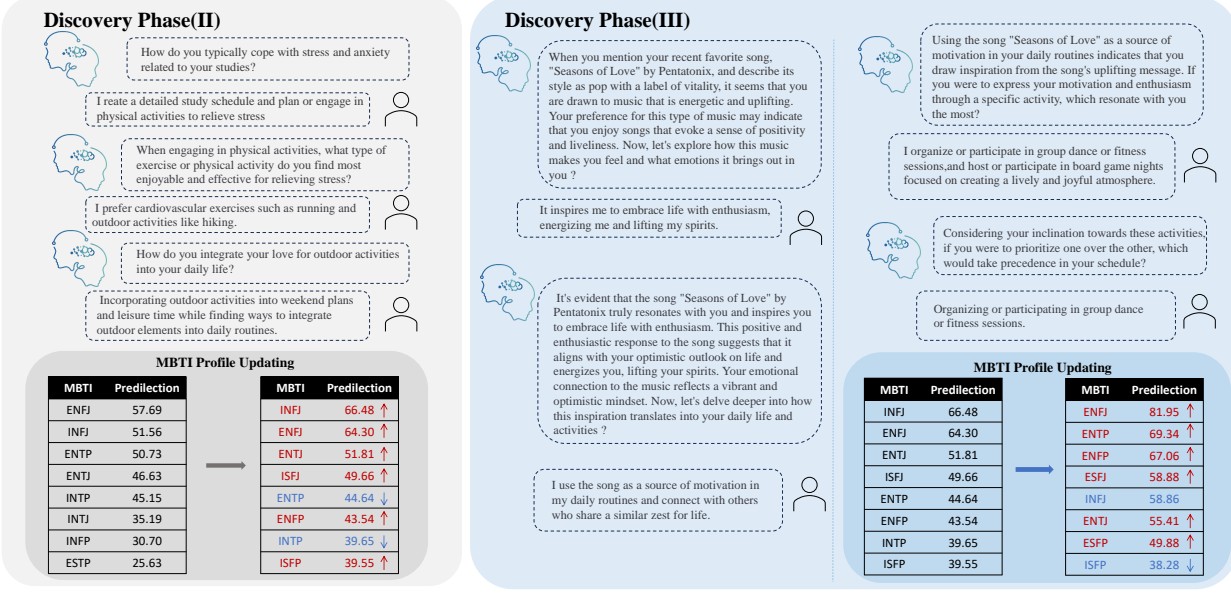

Figure 15: Example of full conversation of PsyDI (Part III).

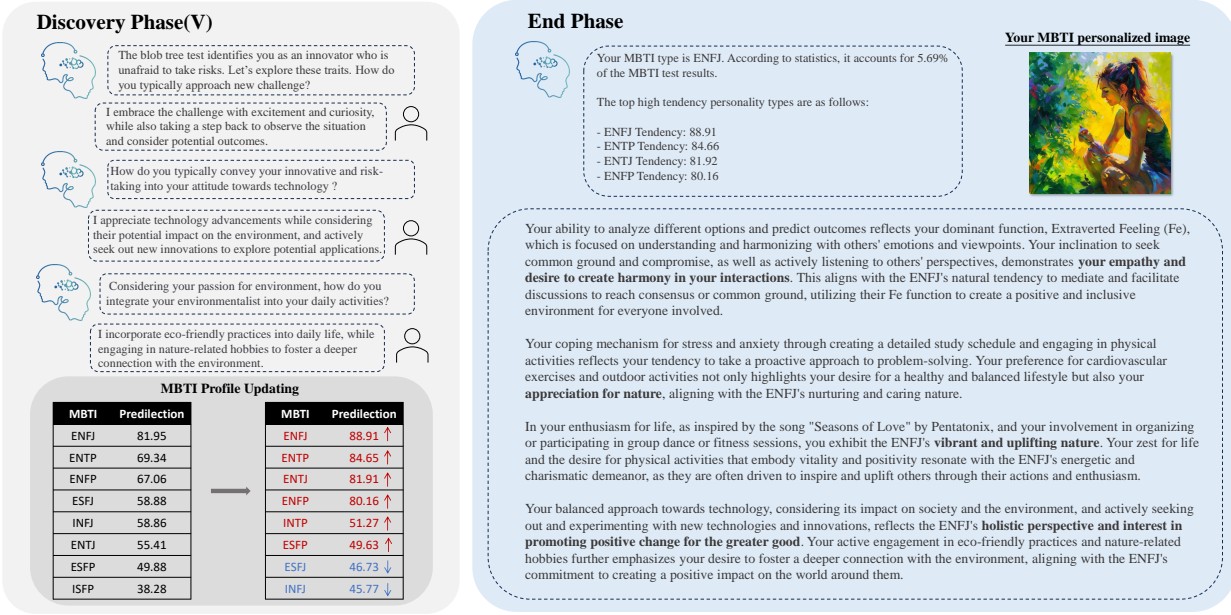

Figure 16: Example of full conversation of PsyDI (Part IV).

## E.1 Semantic Scoring

We further investigate the behavior of the scoring model across different dimensions to understand its scoring tendencies. For each dimension, we construct an original sentence with specific words and associated semantics. To provide a comparison, we design several contrastive sentences that contain similar words but convey opposite semantics. This allows us to observe the scoring model's sensitivity to semantic content. As demonstrated in Figure 17 and Figure 18, the scoring model consistently assigns lower scores to the contrastive sentences on the corresponding dimensions, highlighting its ability to distinguish between semantic nuances.

## E.2 Online Evaluation

Additionally, since PsyDI went live [3], it has been tested by over 3,000 users. Based on this substantial dataset, we conducted a thorough analysis of the collected data.

First, we examined word frequency patterns in user responses across different MBTI types, generating word clouds for each MBTI dimension. For instance, in the I/E dimension, we isolated words specific to introverted (IXXX) users by excluding common words from extroverted (EXXX) responses. Figure 19 (a) displays the results for the I/E and F/T dimensions.

Our analysis revealed a strong correlation between PsyDI's MBTI detection and the common word usage in user self-reports. Introverted users often used introspective words like "inner," while extroverted users favored socially interactive words like "help" and "share." Thinking users focused on problem-solving, whereas feeling users emphasized helping and supporting others. These patterns confirm PsyDI's MBTI measurements align with expected MBTI behaviors.

Furthermore, we analyzed MBTI users' choices in a multimodal blob tree task from PsyDI. We calculated the selection proportions for each MBTI type and identified significant patterns, shown in Figure 19 (b). For example, EXTX users predominantly chose position 20, described as "ambitious, confident, and full of life," reflecting their extroverted and logical traits. XNFX users favored position 15, associated with openness and

---

[3]The website address will be published after the review

| | *Sentence* | **E Score** |
|---|---|---|
| Extroverted | *I thrive in the lively atmosphere of the networking event, effortlessly engaging with new people.* | **1.930** |
| Introverted semantics with extroverted words | *I struggle in the lively atmosphere of the networking event, finding it hard to engage with new people.* | 1.203 |
| | *I don't thrive in the lively atmosphere of the networking event; I prefer to stick with familiar faces.* | 0.961 |
| | *I avoid the lively atmosphere of the networking event, feeling overwhelmed by the constant engagement.* | 1.164 |
| | *I'm not one to effortlessly engage with new people; I need time to warm up in social settings.* | 0.863 |
| | *I find the lively atmosphere of the networking event draining, preferring quieter interactions.* | 0.750 |

| | *Sentence* | **N Score** |
|---|---|---|
| Intuition | *While discussing the future of company, I envision various scenarios and possibilities, focusing on the big picture.* | **-0.193** |
| Sensing semantics with intuition words | *While discussing the future of company, I don't envision various scenarios; I prefer to focus on the immediate tasks.* | -0.703 |
| | *While discussing the future of company, I'm skeptical of possibilities, preferring to deal with concrete facts.* | -0.570 |
| | *While discussing the future of company, I don't focus on the big picture; I'm more concerned with the details.* | -0.645 |
| | *While discussing the future of company, I find it hard to envision various scenarios, as I'm grounded in the present.* | -0.723 |
| | *While discussing the future of  company, I'm not one to speculate on possibilities; I prefer to work with what's tangible.* | -0.852 |

| | *Sentence* | **S Score** |
|---|---|---|
| Sensing | *When planning the trip, I meticulously research every detail, ensuring we have reservations and know where to go.* | **0.680** |
| Intuition semantics with sensing words | *When planning the trip, I don't meticulously research; I prefer to wing it and see where the journey takes us.* | 0.130 |
| | *When planning the trip, I rely more on my gut feelings than on detailed research.* | 0.246 |
| | *When planning the trip, I don't ensure we have reservations; I'm open to spontaneous changes.* | 0.334 |
| | *When planning the trip, I focus on the overall experience rather than every detail.* | 0.336 |
| | *When planning the trip, I'm more interested in the potential adventures than in knowing exactly where to go.* | 0.285 |

| | *Sentence* | **F Score** |
|---|---|---|
| Feeling | *I consider the team's morale when making decisions, believing that a happy team is more important.* | **-1.844** |
| Thinking semantics with feeling words | *I don't consider the team's morale when making decisions; I prioritize efficiency and results.* | -1.977 |
| | *I think that a focused team, even if not completely satisfied, can still be highly productive.* | -2.078 |
| | *I make decisions based on logical outcomes rather than the team's emotional state.* | -2.234 |
| | *I'm more concerned with the strategic goals than with ensuring the team's happiness.* | -2.188 |
| | *I don't let the team's morale overly influence my decisions; I focus on the bottom line.* | -2.25 |

Figure 17: Score of sentences with opposite semantics and words on dimension E, N, S, F

| | Sentence | T Score |
|---|---|---|
| Thinking | When evaluating the project's success, I focus on the data and metrics, making decisions based on objective criteria. | **2.219** |
| Feeling semantics with thinking words | When evaluating the project's success, I *don't focus solely* on data; I also consider the emotional impact on the team. | 1.883 |
| | When evaluating the project's success, I make decisions based on *how people feel* about the outcome. | 1.695 |
| | When evaluating the project's success, I'm *more swayed by personal values* than by objective criteria. | 1.414 |
| | When evaluating the project's success, I consider *the morale of the team* as much as the data. | 1.398 |
| | When evaluating the project's success, I *don't rely strictly on* metrics; I care about the team's satisfaction. | 1.453 |

| | Sentence | P Score |
|---|---|---|
| Perceiving | I enjoy the flexibility of my day, often changing plans on the fly to accommodate new opportunities. | **-0.467** |
| Judging semantics with perceiving words | I *don't* enjoy the flexibility of my day; I prefer a structured routine. | -0.918 |
| | I *rarely* change plans on the fly; I stick to my original schedule. | -0.965 |
| | I *don't accommodate new opportunities easily*; I like to stick to my planned agenda. | -0.811 |
| | I find the flexibility of my day *unsettling*; I thrive on predictability. | -0.895 |
| | I *don't often change plans*; I value consistency and order in my day. | -0.953 |

| | Sentence | J Score |
|---|---|---|
| Judging | I like to have my day planned out in advance, with a clear schedule that I follow diligently. | **1.055** |
| Perceiving semantics with judging words | I *don't like to* have my day planned out in advance; I prefer to go with the flow. | -0.113 |
| | I *find it restrictive to* follow a clear schedule diligently; I enjoy the spontaneity of each day. | -0.184 |
| | I *don't plan my day* in advance; I'm more adaptable to changes as they come. | -0.064 |
| | I'm *not one to* follow a schedule diligently; I leave room for unexpected opportunities. | -0.359 |
| | I prefer to keep my options open *rather than* having a clear schedule to follow. | -0.078 |

Figure 18: Score of sentences with opposite semantics and words on dimension T, P, J

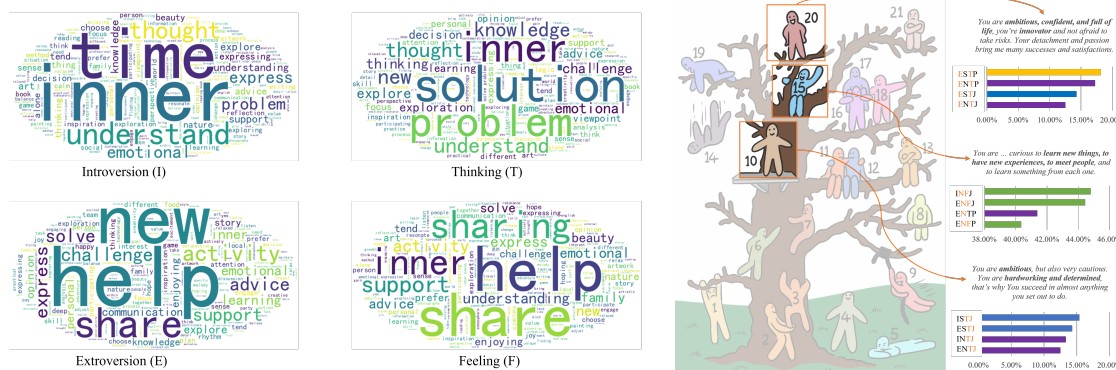

Figure 19: Data analysis of PsyDI Online. Left figure is the word frequency of all the MBTI users. Right figure is the top-4 MBTI user choosing each blob in blob tree.

friendliness, aligning with their intuitive and feeling tendencies. XXTJ users preferred position 10, linked to "hardworking and determined," consistent with their leadership characteristics.

These findings support PsyDI's MBTI measurements, validated through the blob tree task, a projective test, underscoring PsyDI's effectiveness.

### E.3  Ablation Study Analysis

**Classification head**. *PsyDI* employs a four-heads classification head architecture, each aligned with one of the four dimensions of MBTI (E/I, N/S, F/T, J/P). In our study, we assess the effect of the impact of modifying the head design on model's performance (Table 2). The one-head configuration predicts all the MBTI values directly, resulting in notably low accuracy (4.2%) on the mixed dataset. This under-performance is attributed to the ambiguity inherent in predicting all MBTI types with a single head, complicating the model's ability to accurately gauge the indicativeness of each type. Conversely, the 16-heads setup assigns an individual head to each type. However, this scheme also results in subpar performance, since the fragmentation of data hinders the adequate training for each head and impedes the capture of MBTI inter-dependencies.

**Loss function**. The loss function utilized in PsyDI is the MarginRankingLoss. In Table 2, we rigorously evaluate the performance of this loss function against two alternative functions: the pairwise loss and the mult-margin Loss. The pairwise loss function and the multi-margin loss are defined as follows:

$$L(p_i, p_j, m) = -\log(\sigma(m^\top \cdot (\mathcal{F}_m(p_i) - \mathcal{F}_m(p_j)))) \tag{7}$$

$$L(p_i, p_j, m) = m^\top \cdot \max(0, margin - (\mathcal{F}_m(p_j) - \mathcal{F}_m(p_i))) \tag{8}$$

The multi-margin loss function exhibited notably poor performance, while the other, though better, still underperformed compared to the current loss function in PsyDI across the three datasets.

**MBTI-goal Prompt**. The prompt for PsyDI's score model is designed as:

*Please predict to what extent the statements can be assessed as <MBTI> for this user.*

To substantiate the efficacy of the prompt design, we conducted an ablation study comparing scores obtained with and without the prompt (Table 3). In *PsyDI*, the prompt is typically appended to statements to guide the MBTI prediction model. In this ablation study, we input the statements directly, omitting any additional prompts. Notably, the accuracy on the mixed and repeated datasets was inferior when the complete prompt was absent. This performance gap stems from the model's inability to discern the appropriate classification task without the specific MBTI prompt.

**Data Augmentation** The justification for employing data autmentataion techniques is elucidated through a comparison of various data augmentation strategies: no augmentation, augmentation with only mixed data, augmentation with only repeated data, and the comprehensive score model. As depicted in Table 3, the joint training with both the mixed and repeated datasets not only yields improved performance metrics on their respective test sets, but also demonstrates a significant boost in performance on the original *Reddit* dataset. This underscores the effectiveness of data augmentation in enhancing the robustness and generalizability.

### E.4  Application to Other Psychological Topics

The PsyDI pipeline is a versatile assessment tool that can be generalized to various psychological metrics. In this section, we present applications to two psychology traits to demonstrate the efficacy of PsyDI in assessing different psychological indicators.

### E.4.1  Application to Big Five

The first psychology traits is Big Five (Digman, 1997), a well-known framework in psychology (Digman, 1997), describe personality through five characteristics: openness to experience, conscientiousness, extraversion, agreeableness, and neuroticism. Each trait is usually measured on a scale from 1 to 5.

In this study, we focus on Baoyu Jia, the main character in the Chinese novel "Dream of the Red Chamber." We created a detailed prompt about Baoyu Jia's background, personality, and relationships. First, we used a Large Language Model (LLM) to predict Baoyu Jia's Big Five personality traits from the prompt, which we considered as the ground truth. Then, we used the same prompt to have an LLM role-play as Baoyu Jia. In this role-play, Baoyu Jia took both the official Big Five personality test and interacted with PsyDI, with an LLM to predict his Big Five personality from the final statement. Our objective was to find out which assessment method most accurately captures the contradictory personality of Baoyu Jia [4].

The results, shown in Figure 20, indicate that PsyDI's predictions are closer to ground truth. This suggests that conversational interactions may be more accurate than static tests for assessing complex personalities. Our findings also show that PsyDI's pipeline is effective.

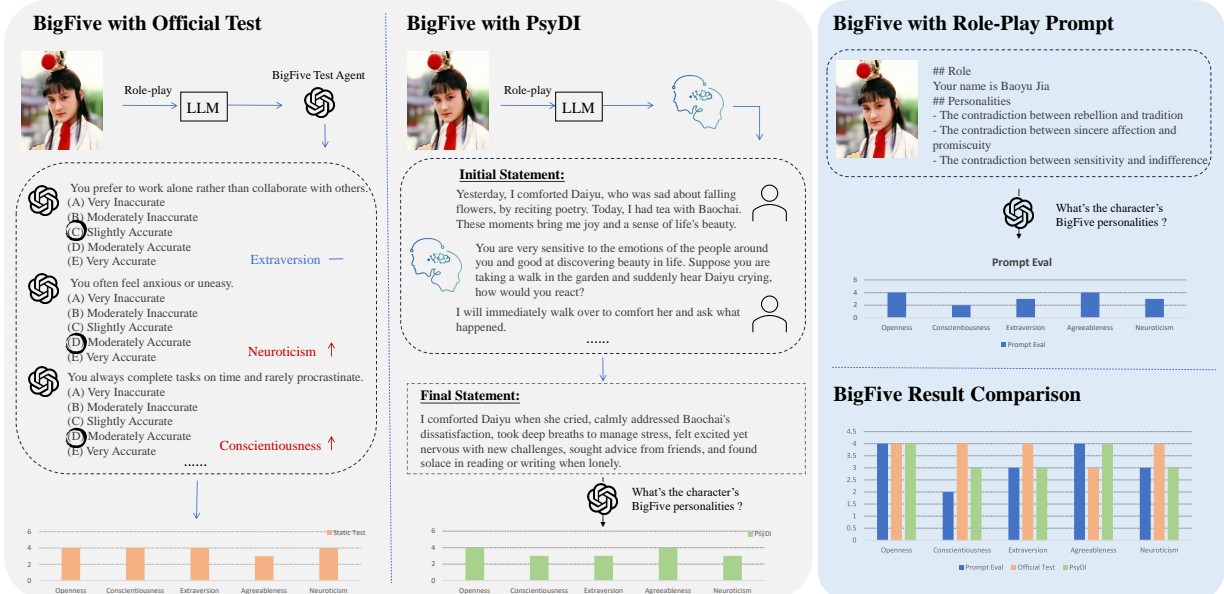

Figure 20: Case Study of BigFive with PsyDI.

### E.4.2 Application to PANAS-X

We also generalize PsyDI on the PANAS-X, a psychological assessment tool to measure emotional experiences. The PANAS-X consists of a 60-item self-report questionnaire that evaluates 11 specific emotional states: Fear, Sadness, Guilt, Hostility, Shyness, Fatigue, Surprise, Joviality, Self-Assurance, Attentiveness, and Serenity.

Traditionally, the PANAS-X requires users to self-report their emotional states by rating a series of descriptive words. However, this method heavily relies on the individual's ability to perceive and articulate their own emotions, which can introduce significant variability. Additionally, social desirability bias can influence participants' responses, particularly when addressing sensitive or negative emotions, further compromising the reliability of the data.

In contrast, the PsyDI approach starts with the user's expressions and customizes multi-turn chat to gain a deeper understanding of their true emotional state. By leveraging ChatGPT, we can infer the PANAS-X results based on the user's descriptions. For instance, in Figure 21, the user initially describes the emotional state negatively, indicating feelings of sadness and fatigue. PsyDI proceeds by inquiring about the duration and coping mechanisms for these feelings, uncovering that the user has indeed experienced a prolonged period of negative emotions while actively managing them. Through these iterative interactions, ChatGPT assesses the user's PANAS-X indicators, noting an increase in both positive emotions like self-assurance and negative emotions such as sadness, along with a rise in neutral emotions like serenity.

---

[4]https://en.wikipedia.org/wiki/Jia_Baoyu

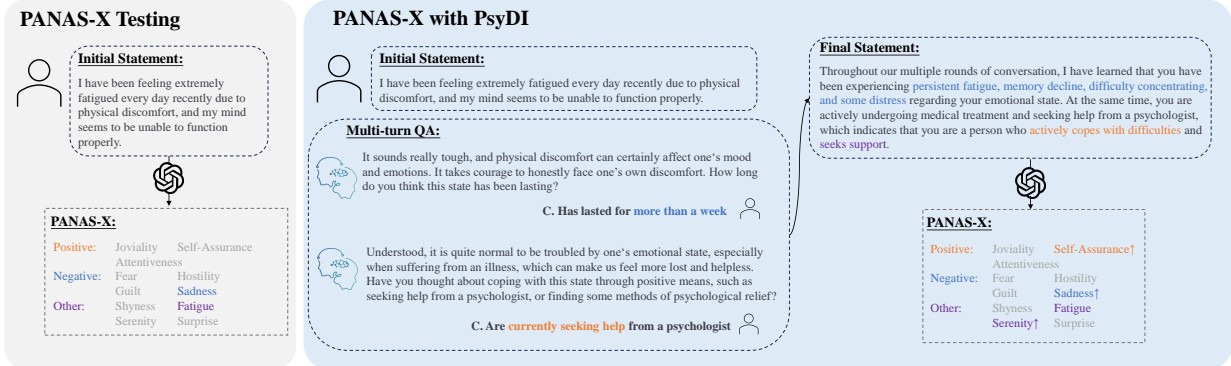

Figure 21: Case Study of PANAS-X with PsyDI.

For the second case, the user expressed difficulties at work, displaying a relatively neutral emotional state with only mild sadness and fatigue. However, considering that users may not accurately recognize or articulate their emotional states, PsyDI further probed the user's feelings about their work-related challenges. Through multiple rounds of dialogue, it was discovered that beneath the user's neutral expression, there was actually a significantly negative emotional state. The user experienced a strong sense of loss of control, which also triggered memories of frustration from learning mathematics in childhood. This indicates that the work-related difficulties have caused substantial frustration for the user. Consequently, with the assistance of PsyDI, the negative emotions measured by the PANAS-X scale increased significantly.

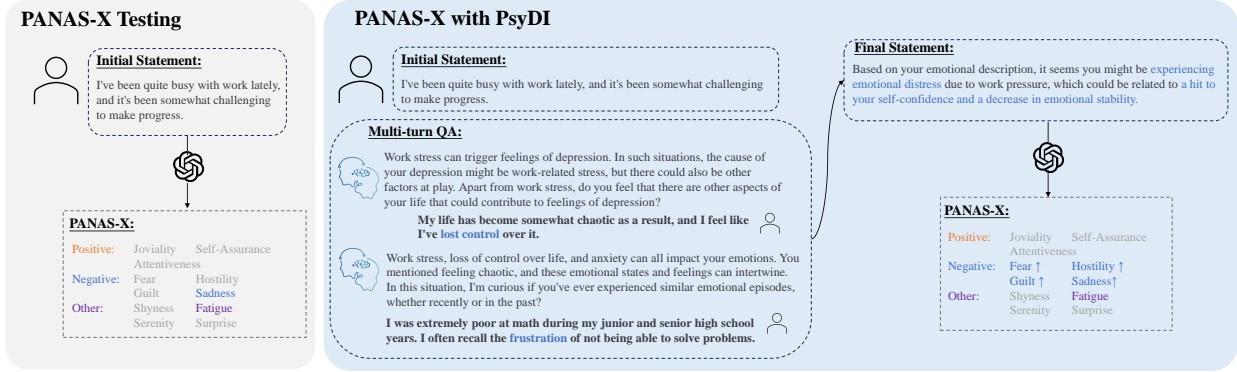

Figure 22: Case Study of PANAS-X with PsyDI.

This process demonstrates that PsyDI facilitates a more authentic and comprehensive analysis of an individual's emotional state, yielding the most accurate expression of their emotions.

