# OpenReview forum: "PsyDI: Towards a Personalized and Progressively In-depth Chatbot for Psychological Measurements"
_TMLR — Rejected by TMLR_

### Review · Reviewer_LNuR · 2024-09-10

**Summary Of Contributions:**

This paper proposes an LLM pipeline for adaptively understanding the MBTI type of a person with the goal of minimizing the number of interactions and maximizing the identification accuracy. Although the area is out of my domain of expertise, I found the paper interesting to read, and appreciate that the authors presented a practical solution for a seemingly difficult problem in psychology.

**Audience:**

Yes

**Claims And Evidence:**

Yes

**Requested Changes:**

I would recommend the authors add additional details of their methods as described above, with an emphasis on how the MDP formulation is used in their pipeline and how their multi-turn chat is implemented. It is possible that I missed other details of the framework so please add the corresponding details as needed.

The authors should include a discussion on the quality of the open-source data and potentially perform robustness checks of their proposed pipeline.

Happy to update the score.

**Strengths And Weaknesses:**

The paper is in general easy to follow. However, there is a slight disconnection between different sections of the paper. Additionally, I found the paper lacks a detailed description of the methods in the main body, hindering the reproducibility of their proposed method.

In Section 2, the authors provided an MDP formulation of the problem. However, when the authors introduce different components of the pipeline in Section 3, it is unclear where the MDP formulation is used to optimize the actions taken by the pipeline.

In Section 3.3, the authors mentioned that a multi-chat structure was implemented to mitigate the limitation posed by LLM being not focused. However, from the description in Section 3.3 and in Appendix C, it is unclear to me how the sequence of actions is determined. Is it determined by the MDP formulation listed in Section 2 or do the authors use a fixed set of rules to determine the action sequence?

Additionally, the paper lacks a description of the quality of the data. Since the experiments are mostly based on open-source data, it would be beneficial for the authors to discuss the quality of the labels in the data set and perform robustness checks. It would be useful to include a literature review of the set of papers that utilized this dataset and acknowledge the limitation of the dataset somewhere in the paper.

---

> ### Author Response · Authors · 2024-11-14
>
> Dear Reviewer,
> Thank you for your detailed comments! We address your concerns in detail below, aiming to clarify our methodology and results.
>
> ## MDP Modeling
> > In Section 2, the authors provided an MDP formulation of the problem. However, when the authors introduce different components of the pipeline in Section 3, it is unclear where the MDP formulation is used to optimize the actions taken by the pipeline.
>
> Regarding your first question, please refer to the **MDP Modeling** part in general response.
>
> ## Action Sequence
> > In Section 3.3, the authors mentioned that a multi-chat structure was implemented to mitigate the limitation posed by LLM being not focused. However, from the description in Section 3.3 and in Appendix C, it is unclear to me how the sequence of actions is determined. Is it determined by the MDP formulation listed in Section 2 or do the authors use a fixed set of rules to determine the action sequence?
>
> We would like to provide further clarification regarding the actions mentioned in Section 2. The actions defined in the MDP framework are conceptual and serve as a basis for the hierarchical decision-making process we employ. Specifically, actual decision-making involves a hierarchical action structure to ensure effective questioning.
>
> Initially, the primary action set {w1,…,wn} is decomposed into two sub-actions: selecting a statement and formulating a question based on that statement. Mathematically, this can be represented as:
>
> $$A'\in A, where A'=N\times\lbrace w_1^{'},...w_n^{'}\rbrace $$
>
> Here, N denotes the set of possible statements, and {w1',w2',…,wn'} represents the set of potential questions.
> Next, with the multi-turn dialogue capabilities of the LLM, we further decompose the questioning process into 3-5 iterative steps. Each step involves selecting a question from the set {w1',w2',…,wn'} based on the current prompt p and the dialogue history history. This can be formally expressed as:
>
> $$A''=\lbrace w_1^{''}, ...,w_n^{''}|p, history\rbrace \in \lbrace w_1^{'},...w_n^{'}\rbrace$$
>
> Thus, our overall action structure can be summarized as:
>
> $$A=N\times\lbrace w_1, ...,w_n|p, history\rbrace$$
>
> This hierarchical approach ensures that each action is both meaningful and context-sensitive, indirectly optimize the reward.
>
> ## Quality of dataset
> > Additionally, the paper lacks a description of the quality of the data. Since the experiments are mostly based on open-source data, it would be beneficial for the authors to discuss the quality of the labels in the data set and perform robustness checks. It would be useful to include a literature review of the set of papers that utilized this dataset and acknowledge the limitation of the dataset somewhere in the paper.
>
> We appreciate your suggestion regarding the discussion of data quality and robustness checks. We have addressed this by providing a detailed description of the dataset construction process and its characteristics in Appendix B.1 in the revised version of paper.
>
> Firstly, the dataset we used, exemplified by the Reddit dataset, has been widely recognized with over 9K downloads. During the dataset construction process, we carefully handled the distribution and quality of the data. Specifically, we filtered out statements based on length and sampled 400 instances per MBTI type, ensuring a balanced and representative dataset. This process is detailed in Appendix B.1 in the revised version of paper, where we also demonstrate the dataset's coverage of common noun-verb combinations relevant to MBTI types. We also conducted experiments on MBTI role-play bot, as well as observing consistent phenomena with PsyDI Online on real-world data. These results substantiate the robustness and reliability of our approach.
>
> The discussion and limitations of the dataset and the whole pipeline is added to Section 7 in our paper.

---

### Review · Reviewer_HcU6 · 2024-09-13

**Summary Of Contributions:**

This paper presents a chatbot PsyDI for psychological assessments (specifically MBTI). It is motivated by the lack of personalization and depth of score-based assessments and the inaccessibility of consultation with a trained professional. The paper distinguishes PsyDI from prior work by its customized, multi-turn interactions, unlike previous single-round chatbots.

The method consists of initializing the model according to the person's existing statements, then selection of statements based on how indicative (a defined metric) the statement is of all of the person's most likely MBTI types, then a multi-turn chat with various structured statements to follow up on the desired statement. It also uses a score model to determine the proxy variables. The LLM's MBTI type predictions for a given statement are checked for correlation with assigned labels, then the accuracy with respect to ground truth is treated as a proxy variable indicating the indicativeness of the statement. The score model is given the ranking of the prediction accuracy rather than the exact value due to the inherent noise of these measurements and LLM predictions.

The paper uses pair-wise datasets to abstract away length and repetition from getting high scores.

Experimental evaluation includes data from Reddit, 16Personalities, and Diamente. It was compared to existing LLMs and outperformed them. Ablations were conducted on the classification heads, loss function, prompt, and data augmentation, and the score model was evaluated as a standalone model. The results show the efficacy of the PsyDI framework on both real humans and simulated MBTI personalities.

**Audience:**

Yes

**Broader Impact Concerns:**

Many. Psychological diagnosis is sensitive, and there is no discussion of the safety, implications, and threats and risks of such a system. These are magnified by the fact that it will be used unsupervised.

The paper also doesn't discuss data protection, user rights, and constraints the researchers might have followed to uphold these.

**Claims And Evidence:**

Yes

**Requested Changes:**

- Better structure in both methods and results section. In methods section, the modules are signposted already, but have the writing flow better going through each part of the model. In the results section, have claim-driven subsubsections within each subsection - tell us the main takeaway as a header, not just buried in text.
- Ideally, a more rigorous framework than MBTI would be shown, e.g. Big Five.
- Show more fleshed-out examples. The appendix is very useful, but I'd want to actually see what the language model is generating in its multi-turn trace, since this is such a core part of the work.
- More extensive human evaluation - ask them more specific questions about the experience, for example

**Strengths And Weaknesses:**

### Strengths
- Well motivated - the choice between accessible but impersonal and personal but inaccessible is a longstanding problem in psychological diagnosis. The proposed solution (and family of such solutions) is promising
- Results speak for themselves - people liked it, which is meaningful
- Methodology is described comprehensively - there's lots of detail. The system is simple in a good way, and each part of it is fleshed out
- Use of proxy variables is clever. At first I was confused as to why the LLM was involved at all, but that was a matter of clarity - it makes sense after reading, and is a thoughtful way to get around the challenge of hand-designing meaningful quantifications of psychological expressions.

### Weaknesses
- MBTI is generally not considered a rigorous psychology framework
- Unclear what a multi-turn conversation looks like. There are some piecemeal examples in the appendix, but a full conversation would be useful; the link in the github crashes after a few questions. It's not clear from section 3 how follow-up statements are determined, and how pre-scripted they are.
- Though methodology is described comprehensively, it's not explained that clearly. Figures 2 and 3 are useful, but they could be integrated into the text more. The text could also be structured more - there are large walls of text explaining technical concepts.

---

> ### Author Response · Authors · 2024-11-14
>
> Dear reviewer,
> Thank you for your insightful comments. We have reviewed each point and have addressed them as follows:
> ## Random Nature of MBTI Test
> > MBTI is generally not considered a rigorous psychology framework
>
> Please refer to the **Limitations of MBTI Test** part in general response.
>
> ## Multi-turn Conversation Details
> > Unclear what a multi-turn conversation looks like. There are some piecemeal examples in the appendix, but a full conversation would be useful; the link in the github crashes after a few questions. It's not clear from section 3 how follow-up statements are determined, and how pre-scripted they are.
>
> We appreciate your suggestion to clarify the multi-turn conversation process. To provide a comprehensive view of the multi-turn conversation and the change of the MBTI profile, we have included a full example of the PsyDI full conversation in Appendix D.2 in the revised version of paper. This example demonstrates the interaction flow and how the MBTI profile changes over the course of the conversation.
>
> We apologize for the inconvenience caused by the crashing link on our GitHub. This issue arises due to server capacity limitations during high traffic periods. We encourage you to try accessing the link again, as we have now added both Chinese and English versions with separate entries.
>
> Regarding the generation of follow-up statements, after 3-5 rounds of dialogue, PsyDI summarizes the user's information and presents it in the first person. For instance, if the initial statement is "I feel anxious," after discussing work-related topics, the follow-up statement might be "I feel anxious about work because I prioritize work outcomes..." This detailed description reflects the user's thoughts and preferences.
>
> ## Clarity and Structure in Methodology Section
> > Though methodology is described comprehensively, it's not explained that clearly. Figures 2 and 3 are useful, but they could be integrated into the text more. The text could also be structured more - there are large walls of text explaining technical concepts.
>
> We appreciate your feedback on the clarity and structure of our description. To enhance readability and comprehension, we have integrated Figures 2 and 3 into the text, providing detailed explanations in Sections 3 and 4. Additionally, we have restructured the text to provide a clearer flow of information.
>
> ## Safety, Ethical, and Data Protection Concerns
> > Many. Psychological diagnosis is sensitive, and there is no discussion of the safety, implications, and threats and risks of such a system. These are magnified by the fact that it will be used unsupervised.
>
> The paper also doesn't discuss data protection, user rights, and constraints the researchers might have followed to uphold these.
> We appreciate your concerns regarding the safety, implications, and risks associated with psychological diagnosis. To address these issues, we have implemented defensive mechanisms in our prompts to ensure that the LLM does not generate harmful content, as shown in Appendix C. Specifically, if a user's response deviates from the intended topic, the LLM is prompted to refuse further interaction and ask the user to return to the topic.
>
> Furthermore, we have added the discussion and limitations in Section 7 to point out potential issues. This section provides a comprehensive overview of the ethical considerations of our pipeline.

---

> ### Author Response · Authors · 2024-12-11
>
> Dear Reviewer HcU6,
>
> We hope you are doing well. We are writing to kindly follow up on the rebuttal for our manuscript "PsyDI: Towards a Personalized and Progressively In-depth Chatbot for Psychological Measurements" (Submission Number: 3184). If there are any further questions or clarifications needed, please let us know.
>
> Thank you for your time and consideration.
>
> Best regards,
> Authors

---

### Review · Reviewer_Knck · 2024-11-04

**Summary Of Contributions:**

This paper proposes a LLM-based framework PsyDI for prediction of users’ MBTI types. This framework facilitates multiple rounds of multi-turn chats with the user, each round guided by a selected statement that will help clarify the MBTI type of the user. The goal of this framework, in comparison to standard/traditional methods, is to be more personalized and engaging. The experiment results validate separately subset(elements) of the framework and the final MBTI-type prediction accuracy using Chatbots from c.ai as ground truth.

**Audience:**

Yes

**Claims And Evidence:**

No

**Requested Changes:**

See points in <Weaknesses> in the above section.

**Strengths And Weaknesses:**

Strengths:
1. This paper contains clear and informative visualizations of both the composition of the PsyDI pipeline as well as experiment results.
2. The authors conduct experiments as attempts to examine and validate subset of the pipeline, e.g. the score model, to increase confidence in the modelling choice.

Weaknesses:
The fundamental concern is that the goal and the approach in this paper lack scientific rigor. My main concerns can be summarized in the first two points below.
1. Invalidity and the random nature of the Myers–Briggs test as a psychological assessment: Despite its popularity, various research in psychology have stated cautions and criticisms against this test, including but not limited to test-retest reliability; empirical validity; motivational distortion, etc., which very much limits the actual utility of the MBTI. See references [1,2] and the many references therein. Without further evidence, I am unsure whether the goal of predicting the unstable-in-nature (and with questionable utility) MBTI profile of users is useful and of interest. Given the substantial concerns raised by researchers that warrant against its professional use in psychology, the scope of “Chatbots for Psychological Measurements” is also unjustified.
2. In Section 5, chatbots from c.ai are used as a proxy for human users and the “MBTI” settings of those chatbots are used as ground truth for validating the accuracy of PsyDI predictions. My concern in the use of this ground truth, is with replacing human subjects with LLMs in participating such tests. For example, [3] and references therein provide concrete evidence that LLMs deviate from humans in their answers in face of personality tests. Without further analysis (e.g. a detailed examination of the difference between chatbot answer distributions and human answer distributions), the use of c.ai bots in replacement of human is questionable.

------

Besides the two high-level main concern above, I also found that various parts in the paper lacks clarity. Please see the points below.

3. The modeling choice of MDP (Section 2) seems arbitrary and unhelpful. Are any algorithms used that were designed to optimize this MDP? Was this formulation later used in any other places in the paper?
4. I am lost at the definition of the MBTI profiles. One on hand, it contains “individual score for each trait” and appears to be a 4-dimensional vector, on the other hand, authors claim that (for a fixed user with statement history), they maintain a score for each MBTI type, which makes the MBTI profile a 16-dimensional vector. --> Can the authors explain (1) how the MBTI profile is defined, and (2) how it leads to the final prediction which appears to be a 16-dimensional vector whose elements are probabilities that sum up to 1.
5. What are the initial statements from the user?
6. In eq(5), if M_j^i and M_k^i are only used in the XOR operation, why do they need to be “counterpart” of m?
7. In Section 3.3, how do the authors make sure that the LLMs can comply with the prompt throughout the multi-turn chat? Has this been verified?
8. In Section 1, authors listed several principles that PsyDI is designed to follow. Can authors clarify how PsyDI (presumably in the multi-turn chat part) follow this principle? “highly dependent on the user context, including cultural, linguistic, and individual factors to minimize the biased impact.”

References
1. Boyle, Gregory J. "Myers‐Briggs type indicator (MBTI): some psychometric limitations." Australian Psychologist (1995).
2. Pittenger, David J. "Cautionary comments regarding the Myers-Briggs type indicator." Consulting Psychology Journal: Practice and Research (2005).
3. Sühr, Tom, et al. "Challenging the Validity of Personality Tests for Large Language Models." arXiv e-prints (2023).

---

> ### Author Response · Authors · 2024-11-14
>
> Dear Reviewer,
> We would like to express our gratitude for your thorough review and valuable feedback on our manuscript. We have carefully considered each comment and have addressed them in detail below.
> ## Random Nature of MBTI Test
> > Invalidity and the random nature of the Myers–Briggs test as a psychological assessment: Despite its popularity, various research in psychology have stated cautions and criticisms against this test, including but not limited to ...
>
> Regarding your point about the invalidity and random nature of the MBTI as a psychological assessment, please refer to the **Limitations of MBTI Test** part in general response.
>
> ## Accuracy of chatbots from c.ai
> > In Section 5, chatbots from c.ai are used as a proxy for human users and the “MBTI” settings of those chatbots are used as ground truth for validating the accuracy of PsyDI predictions...
>
> Regarding the issue you mentioned about the deviation between chatbots and human responses as testers, we would like to clarify that the bot we use is not applicable to the conclusions drawn in the paper you referenced. The role-model prompts in the paper you cited consist of two to three sentences describing a person in the PersonChat dataset. In contrast, the bot we use represents a typical personality, with prompts that inherently contain highly relevant information about personality, resulting in responses that are relatively more typical.
>
> Additionally, the prompt design for c.ai is modular, incorporating amount of information about a specific character. This type of prompt differs significantly in character-building effectiveness from simple descriptive prompts.
>
> To demonstrate that the c.ai box we use can reflect the characteristics of a specific MBTI group, we have added Figure 10 in Appendix B.4 of the revised version of our paper. In the PsyDI process, user will take some questions with four responses under fixed themes, each corresponding to the SJ/SP/NT/NF MBTI personality temperaments. Therefore, we can observe whether c.ai bots and humans align with their respective MBTI types when answering these questions. The results show that c.ai bots and humans are evenly matched for accuracy, and sometimes c.ai bots are even more typical than humans. This further validates the effectiveness of using c.ai bots as typical personality testers.
>
> ## MDP Modeling
> > The modeling choice of MDP (Section 2) seems arbitrary and unhelpful. Are any algorithms used that were designed to optimize this MDP? Was this formulation later used in any other places in the paper?
>
> Regarding the MDP modeling choice in Section 2, please refer to the **MDP Modeling** part in general response. The MDP serves as a conceptual framework for the psychological measurement problem space, not an optimization algorithm. Section 2 elaborates on this, and Section 3 details how heuristic algorithms optimize psychological metric predictions within this framework. The MDP helps define the state space, action space,  transitions function and reward function, guiding the heuristic methods to improve prediction accuracy.

---

> > ### Author Response · Authors · 2024-11-14
> >
> > ## MBTI Profile definition
> > > I am lost at the definition of the MBTI profiles... (1) how the MBTI profile is defined, and (2) how it leads to the final prediction which appears to be a 16-dimensional vector whose elements are probabilities that sum up to 1.
> > Regarding the MBTI profile, it is indeed represented as a 16-dimensional vector, where each element corresponds to the predilection score for one of the 16 MBTI types. The score is a numerical value ranging from 0 to 100, which quantifies the predilection of a given statement to align with a specific MBTI type relative to all other statements. Importantly, these predilection scores are not constrained to sum to 1; they are independently calculated values reflecting the relative strength of each MBTI type's association with the statement.
> >
> > When output the MBTI profile, our scoring model operates as follows:
> >
> > 1. **Initial Scoring**: Each statement is scored for each trait (Extraversion/Introversion, Sensing/Intuition, Thinking/Feeling, Judging/Perceiving) and each MBTI type by the score model, resulting in a 16x4 matrix. Here, 16 represents the 16 MBTI types, and 4 represents each trait within the MBTI. This matrix captures the raw scores for each trait within each MBTI type.
> > 2. **Aggregation and Normalization**: For each MBTI type, the raw scores across the four traits are summed. These sums are then normalized by calculating the percentile rank relative to the sum of scores across all statements. This normalization step ensures that the predilection scores reflect the relative likelihood of each MBTI type, given the entire set of statements.
> > 3. **Final Prediction Vector**: The final prediction is a 16-dimensional vector where each element represents the normalized predilection score for each MBTI type. These scores, while not summing to 1, provide a comprehensive measure of the likelihood of each MBTI type based on the observed statements.
> >
> > In summary, the MBTI profile is a 16-dimensional vector of predilection scores, each reflecting the relative propensity of a statement to align with a specific MBTI type. The scoring mechanism involves a matrix of raw scores, aggregation across traits, and percentile-based normalization to derive the final predilection scores.
> >
> > ## Initial Statement Definition
> > > What are the initial statements from the user?
> >
> > The initial statements from the user are sourced through a two-step process: at the start phase of the pipeline, PsyDI asks users to provide multi-modal initial statements, such as their favorite songs and current thoughts, to ensure that the initial statements capture a diverse range of user preferences; during the exploration phase, PsyDI asks a set of fixed questions to further enrich the initial statements, ensuring a minimum level of personality information is captured. For detailed examples of the whole process, please refer to the explanation in Appendix D.1 and the complete test procedure in Appendix D.2 in the revised version of paper.
> >
> > ## Counterpart Definition in XOR
> > > In eq(5), if M_j^i and M_k^i are only used in the XOR operation, why do they need to be “counterpart” of m?
> >
> > M_j^i and M_k^i are used to represent the binary counterparts of m. For instance, if m denotes the MBTI type INTP, M would be its binary representation [1,1,0,0], where 1 indicates Introverted and 0 indicates Extroverted, with the subsequent dimensions similarly defined. Such binary encoding allows for a systematic and consistent representation of MBTI types across various mathematical operations, which is also used in the sum up operation in the score model.
> >
> > ## LLM Compliance in Multi-Turn Chat
> > > In Section 3.3, how do the authors make sure that the LLMs can comply with the prompt throughout the multi-turn chat? Has this been verified?'
> >
> > The ability of LLMs to follow instructions and advance the conversation with users has been demonstrated in various studies. For instance, the follow-up task in [1] examines the model's proficiency in responding to queries that build upon its preceding responses. GPT-3.5, the LLM used in PsyDI, has shown proficiency in this task. To ensure the conversation module progresses towards the desired goal (testing the user's MBTI), PsyDI employs a meticulously designed prompt structure, detailed in Appendix C. This includes structured prompts and multiple constraints to keep the LLM's dialogue focused on MBTI discussions.

---

> > > ### Author Response · Authors · 2024-11-14
> > >
> > > ## The Principle of Minimizing Biased Impact
> > > > In Section 1, authors listed several principles that PsyDI is designed to follow. Can authors clarify how PsyDI (presumably in the multi-turn chat part) follow this principle? “highly dependent on the user context, including cultural, linguistic, and individual factors to minimize the biased impact.”
> > >
> > > As previously discussed, some of the initial statements are derived from the users' personalized thoughts. And subsequent dialogues revolve around these initial statements, ensuring that the conversation is relevant to the users' life. For instance, a user experiencing work-related anxiety is likely to discuss job performance, while a user concerned about school relationships would not be asked about preferred work style. This approach aligns with the findings in [2], which highlight the impact of the therapist's chosen topics on the therapeutic process.
> > >
> > > Additionally, to mitigate potential biases from overly personalized questions, we have incorporated standardized questions during the initial statement collection to serve as anchors, reducing the risk of biased impacts. Specifically, these questions are randomly selected based on themes, and options are provided based on MBTI temperaments (SP/SJ/NP/NT). The options are designed to have significant separation and are representative of their corresponding MBTI types.
> > >
> > > [1] Kwan, Wai-Chung, et al. "MT-Eval: A Multi-Turn Capabilities Evaluation Benchmark for Large Language Models." arXiv preprint arXiv:2401.16745 (2024).
> > >
> > > [2] Hill, C. E., Corbett, M. M., Kanitz, B., Rios, P., Lightsey, R., & Gomez, M. (1992). Client behavior in counseling and therapy sessions: Development of a pantheoretical measure. Journal of Counseling Psychology, 39(4), 539–549. https://doi.org/10.1037/0022-0167.39.4.539

---

> ### Author Response · Authors · 2024-12-11
>
> Dear Reviewer Knck,
>
> We hope you are doing well. We are writing to kindly follow up on the rebuttal for our manuscript "PsyDI: Towards a Personalized and Progressively In-depth Chatbot for Psychological Measurements" (Submission Number: 3184). If there are any further questions or clarifications needed, please let us know.
>
> Thank you for your time and consideration.
>
> Best regards,
> Authors

---

### Author Response · Authors · 2024-11-14
**General Response**

We would like to express our sincere gratitude to all the reviewers for their valuable and insightful comments. We have addressed common concerns and specific feedback in our detailed responses below. We have made several significant additions and improvements to the document:
1. Section 2: Added an explanation of the relationship between MDP and real-world environment, to illustrate the motivation behind our proposed heuristic algorithm.
2. Appendix B.1: Included a description of the dataset distribution, to illustrate the quality of our training dataset.
3. Appendix B.3: Expanded on the discussion regarding whether MBTI bots can replace real humans in testing, to justify the rationale behind our experimental setup.
4. Appendix D.1: Clarified the correspondence between the stages of PsyDI Online and the phases of the PsyDI pipeline.
5. Appendix D.2: Added a full conversation example of PsyDI Online to illustrate the design of the multi-turn chat interaction.
6. Appendix E4.1: Introduced a test of applying the PsyDI pipeline to the Big Five personality traits, to demonstrate the versatility and applicability of PsyDI in other psychological measurements.
7. Overall: Optimized the descriptions throughout the entire document, and supplemented the motivations in Sections 3 and 4 to enhance the logical flow.
All modifications have been highlighted in blue.

---

### Decision · Action_Editor_eXE9 · 2024-12-27

**Recommendation:** Reject

**Comment:**

This paper introduces PsyDI, a personalized chatbot for psychological measurements. The application was demonstrated via the MBTI framework. While the reviewers found the proposed study to be straightforward and reasonable, they also raised significant concerns regarding novelty (a relatively minor concern) and scientific rigor (a more major concern). Specifically, the reviewers found the choice of MBTI tests to remain unconvincing, with one of the reviewers explicitly voting for rejection and two other reviewers feeling rather neutral.

Quoting the comments from Reviewer Knck in their final recommendation:
> However, evidences such as these are not sufficient in addressing the cautions and criticism against MBTI for being a valid psychological assessment. Hence my concern remains on the paper’s scope on chatbots for "Psychological Measurements”. It is also not clear to me how changing the static scales to the interactive question-answering framework in PsyDI can “mitigating issues related to test-retest instability” (one of the concerns towards MBTI, along with others like motivational distortion) without experimental evidence

>  I feel that a comparison of accuracy (which does not represent the distribution in answers) is not sufficient evidence to clear the potential problems in using c.ai chatbots to replace humans to take personality traits, especially when the accuracy is also provided by LLMs that can have biases of their own. What’s perhaps more helpful would be analysis on the difference in distribution/structure of the answers as in [3] (reference in my original review)

Based on the above considerations, we are not able to recommend acceptance at the current stage, but the authors are encouraged to revise the manuscript for resubmission by incorporating the feedback from the reviewers.

**Audience:**

Yes

**Claims And Evidence:**

The evidence could be strengthened to make the claims convincing.

**Resubmission Of Major Revision:**

The authors may consider submitting a major revision at a later time.